# Research on the Quality Variation Patterns During the Fermentation Process of Coffee-Grounds Craft Beer

**DOI:** 10.3390/foods14061014

**Published:** 2025-03-17

**Authors:** Jiashun Jiang, Jingan Yang, Tong Zhu, Yongjin Hu, Hong Li, Lijing Liu

**Affiliations:** 1Yunnan Coffee Modern Industry College, Yunnan Agricultural University, Fengyun Road 452, Kunming 650201, China; jashunjiang@163.com (J.J.); yangjingan_1874@163.com (J.Y.); tongzhu@ynau.edu.cn (T.Z.); huyjyn@126.com (Y.H.); ynveg@163.com (H.L.); 2College of Food Science and Technology, Yunnan Agricultural University, Fengyuan Road 452, Kunming 650201, China; 3Yunnan Key Laboratory of Coffee, Yunnan Agricultural University, Fengyuan Road 452, Kunming 650201, China

**Keywords:** coffee-grounds craft beer (CGB), physicochemical index, flavor substance, sensory analysis, HS-SPME-GC/MS

## Abstract

To investigate the metabolic differences and mechanisms during the fermentation process of coffee-grounds craft beer, HS-SPME-GC/MS untargeted metabolomics technology was used to study the metabolic differences during the fermentation process of coffee-grounds craft beer. Multivariate statistical analysis and pathway analysis were combined to screen for significantly different metabolites with variable weight values of VIP ≥ 1 and *p* < 0.05. The results indicate that at time points T7, T14, T21, and T28, a total of 183 differential metabolites were detected during the four fermentation days, with 86 metabolites showing significant differences. Its content composition is mainly composed of lipids and lipid-like molecules, organic oxygen compounds, and benzoids, accounting for 63.64% of the total differential metabolites. KEGG enrichment analysis of differentially expressed metabolites showed a total of 35 metabolic pathways. The top 20 metabolic pathways were screened based on the corrected *p*-value, and the significantly differentially expressed metabolites were mainly enriched in pathways such as protein digestion and absorption, glycosaminoglycan biosynthesis heparan sulfate/heparin, and benzoxazinoid biosynthesis. The different metabolic mechanisms during the fermentation process of coffee-grounds craft beer reveal the quality changes during the fermentation process, providing theoretical basis for improving the quality of coffee-grounds craft beer and having important theoretical and practical significance for improving the quality evaluation system of coffee-grounds craft beer.

## 1. Introduction

Coffee is one of the most widely consumed beverages worldwide, and the issue of handling coffee grounds as a major byproduct is becoming increasingly prominent. After the coffee is brewed, the remaining solid residue is called spent coffee grounds (SCG) [1]. SCG contain 39.10% hemicellulose, 23.90% lignin, 17.44% protein, 12.40% cellulose, and 2.29% oil [2], and are rich in active substances such as caffeine, chlorogenic acid, polyphenols, and cucurbitacin [3], providing a unique material basis for the development of functional foods. As a by-product, SCG can be used for high-value and biotechnology innovation, aiming to transform this agricultural waste into high-value-added products through biotechnology and relieve environmental pressure at the same time [4].

According to statistics, the annual consumption of coffee exceeds 10 million tons, inevitably generating about 6 million tons of SCG annually [5]. For every ton of coffee beans produced, about 650 kg of coffee grounds are generated, of which less than 20% are recycled, albeit inefficiently, and the rest are mostly disposed of through landfill or incineration. One sustainable approach involves extracting active ingredients from coffee grounds and converting them into high-value products [6]. Van-Truc Nguyen et al. [7] demonstrated the extraction of biochar from coffee grounds for the adsorption of norfloxacin in aqueous solutions. Similarly, Ibtissam Bouhzam et al. [8] investigated various methods for extracting chlorogenic acid and caffeine from spent coffee grounds. However, most existing research has focused predominantly on the extraction or simple compounding of individual components, whereas the application of coffee grounds in the food sector, particularly the mechanisms underlying their integration into fermented food systems, requires a more comprehensive analysis.

Several researchers have explored the potential applications of SCG in food products. Nuria Martinez-Saez et al. [9] incorporated SCG as a fiber-rich functional ingredient in cookie formulations supplemented with non-nutritive sweeteners (stevia and maltitol) and the prebiotic oligofructose, resulting in products with significantly enhanced nutritional profiles and sensory characteristics. Similarly, Mitra Ahanchi et al. [10] conducted systematic investigations on the effects of varying SCG incorporation percentages (ranging from 0.05% to 30%) on the physicochemical properties and sensory attributes of bakery products and pasta. Their findings demonstrated that strategic integration of SCG into these food matrices offers multifaceted benefits, including enhanced nutritional value, improved sustainability metrics, and favorable sensory characteristics. Additionally, antioxidant phenolic compounds extracted from coffee grounds exhibit notable health benefits, presenting extensive application potential in both the food and pharmaceutical industries [11]. However, the specific mechanisms of these interactions, such as the release and transformation of bioactive compounds during fermentation, are still poorly understood. Further studies are needed to elucidate how the chemical composition of SCG contributes to the metabolic regulation and organoleptic properties of beer in order to optimize its application in functional craft beer production. HS-SPME-GC/MS is used to analyze coffee-grounds craft beer mainly because of its unique advantages in flavor studies. It can comprehensively reveal the changes of volatile compounds during the fermentation process and provide a scientific basis for quality control, process optimization, and product innovation.

The craft beer industry has experienced substantial growth in recent years, prompting manufacturers to seek competitive differentiation through innovative raw material incorporation. Previous research has demonstrated the viability of this approach. For instance, Milinčić, D.D. et al. [12] successfully integrated Prokupac grape pomace seed powder into beer formulations, resulting in products with elevated phenolic content and enhanced sensory characteristics. In a comparable investigation, Nunes Filho, R.C. [13] systematically evaluated the impact of turmeric, black pepper, and aromatic hop varieties on brewing parameters and product stability. SCG represent a promising candidate for such applications due to their complex phytochemical composition, particularly their significant content of bioactive compounds, including caffeine, chlorogenic acid, and related polyphenols. These constituents may undergo various biochemical transformations during fermentation, potentially establishing synergistic interactions that could significantly modulate both the organoleptic profile and functional properties of the resultant beer. Despite these promising attributes, the fundamental mechanisms governing these interactions remain insufficiently characterized, particularly regarding the kinetics of bioactive compound release, their biotransformation pathways during fermentation, and their ultimate impact on product quality parameters. This knowledge gap necessitates further systematic investigation to fully elucidate these complex biochemical processes.

Such further systematic investigations are essential to elucidate the precise mechanisms by which the chemical constituents of spent coffee grounds (SCG) influence metabolic regulation and organoleptic properties during beer fermentation, thereby facilitating the optimization of SCG valorization in functional craft beer production. Headspace solid-phase microextraction coupled with gas chromatography–mass spectrometry (HS-SPME-GC/MS) represents an analytical methodology of choice for characterizing SCG-infused craft beer, owing to its superior sensitivity, reproducibility, and non-destructive sampling capabilities in volatile compound analysis. This analytical approach enables comprehensive characterization of volatile compound evolution throughout the fermentation continuum, establishing a robust scientific foundation for quality assurance protocols, process parameter optimization, and product development initiatives.

The present investigation employs a multifaceted experimental approach to examine the application of SCG in craft beer (CGB) production, integrating comprehensive physicochemical characterization methodologies, time-course fermentation experiments (spanning 7–28 days), and HS-SPME-GC/MS analytical profiling to systematically elucidate compound release kinetics, metabolic regulatory networks, and flavor formation mechanisms within the SCG-supplemented fermentation matrix. The findings from this research will illuminate the complex synergistic interactions between SCG bioactive compounds and beer fermentation processes, providing a theoretical framework for process optimization of functional craft beer manufacturing while concurrently establishing novel valorization pathways for food industry by-products. This research aims to transcend conventional waste management paradigms, providing both fundamental insights and practical methodologies to facilitate sustainable resource utilization and support environmentally responsible transformation within the food industry sector.

## 2. Materials and Methods

### 2.1. Preparation of Samples

The SCG came from Tongsantai Agricultural Development Co., Ltd. in Tengchong City, Yunnan Province, China. The SCG used in the experiment: coffee beans of the variety Catimor were moderately roasted (190–200 °C), ground into a powder with particle diameters of 0.4–0.6 mm, and then concentrated and extracted by an espresso machine to obtain coffee grounds. They were placed in a 60 °C blast drying oven and dried for 48 h. Wheat malt, Australian malt, hops, and Weiss yeast were all purchased from Diboshi Self Brewing Machine Co., Ltd. in Yantai, Shandong Province, China. We placed the beer saccharification solution in a fermentation tank at 20 °C for fermentation, with a fermentation period of 28 days. Samples were taken every seven days and denoted as T7, T14, T21, and T28. Four sets of data were taken, each with three parallel samples, for a total of 12 samples.

### 2.2. Process Flow

The process flow is shown in Figure 1. Specifically, the malt mixture, comprising Australian malt and wheat malt at a ratio of 5:2 (*w*/*w*), was accurately weighed, thoroughly homogenized, and subsequently milled for 3 min using a laboratory-scale grinder (Yongkang Hongtaiyang Electromechanical Co., Ltd. Model: 1000A, Jinhua, China). Coffee grounds (40 mesh) were added with a total malt content of 20%, amylase with a total malt content of 0.4%, and a material-to-water ratio of 1:4. The initial saccharification temperature was 50 °C, and the insulation time was 20 min; we then raised the temperature to 52 °C and held for 40 min; we continued to heat up to 65 °C and maintained for 70 min, continued raising the temperature of the mash to 72 °C, and kept it warm for 10 min; finally, we raised the temperature of the mash to 78 °C. During the heating process, stirring was begun to ensure even heating of the mash, with an average heating rate of 1 °C/min. We directly poured the wort into the filtration tank without insulation for wort filtration, heated and boiled the filtered wort, keeping it boiling for 60 min, and added hops in three different time intervals. The amount of hops added was 0.15% of the total amount of wort. When the wort had boiled for 5 min, we added 30% hops, and, at the 30th min of boiling, we added 40% hops; at the 55th min of boiling, we added 30% hops; and at the 60th min ended the heating process. We poured it into the rotary sedimentation tank for precipitation and, after 20 min, removed the hot solidified material. A plate heat exchanger was used to cool the wort to around 20 °C before entering the fermentation tank. Activated yeast was added at a dosage of 1 ‰ of the total amount of wort after boiling, and we controlled the fermentation temperature at 20 °C. The content of diacetyl represents whether the beer is mature. When the concentration of diacetyl is less than 0.1 mg/L and remains stable after 28 days of fermentation, it indicates that the fermentation is complete.

### 2.3. Physical and Chemical Determination

Alcohol content [14]: measure 100 mL of the sample, add 50 mL of water, heat, collect the distillate, and measure it using the density bottle method. Total acids [15]: total acids is measured using acid-base indicator titration method. Conduct physical and chemical measurements of bitterness units, original gravity, and real degree of fermentation according to Chinese national standards [16]. Bitterness units: transfer 10 mL of un-degassed sample (at 10 °C) to a centrifuge tube. Add one drop of octanol, 1 mL of hydrochloric acid solution, and 20 mL of isooctane. Shake vigorously for 15 min, centrifuge for 10 min, then aspirate the upper layer. Measure the absorbance at 275 nm wavelength. Original gravity and the real degree of fermentation: after alcohol distillation, dilute the residue with water to 100 g. Measure the relative density using a pycnometer to obtain the real concentration, then calculate the corresponding values. pH: determine directly using a pH meter. Chroma: use the colorimeter to measure the values of L*, a*, and b* and calculate them using the following formula:$$\Delta E=\sqrt[{2}]{{\left(L*\right)}^{2}+{\left(a*\right)}^{2}+{\left(b*\right)}^{2}}$$

For protein content, use the BCA protein assay kit method. Calculate protein concentrations based on the standard curve [14]. Draw a standard curve with protein concentration (mg/mL) as the x-axis and absorbance as the y-axis and obtain the linear regression equation y = 0.9232x + 0.0011, R^2^ = 0.9971. Each physical and chemical index was repeated three times.

### 2.4. Determination of Chlorogenic Acid and Caffeine

Make slight modifications according to the method of Yutong Zhang et al. [15]. Accurately weigh 5.0 mg of chlorogenic acid standard and place it in a 50 mL volumetric flask. Dissolve it in anhydrous ethanol and make up to the mark. Then draw up 0, 0.2, 0.4, 0.6, 0.8, and 1.0 mL, dilute them with anhydrous ethanol, and make up to 10 mL, respectively, to obtain chlorogenic acid standard solutions with concentrations of 0, 2, 4, 6, 8, and 10 mg/L. Measure the absorbance at a wavelength of 328 nm using anhydrous ethanol as a blank control. Using the concentration of chlorogenic acid standard solution (mg/L) as the horizontal axis and absorbance as the vertical axis, draw a standard curve and obtain the linear regression equation y = 0.0288x + 0.1565, R^2^ = 0.9952.

For determination of caffeine according to Chinese national standard [17], weigh 5 g of the sample, dilute to 5 mL with water, and mix thoroughly. Add 0.5 g of magnesium oxide, shake vigorously, and allow the mixture to settle. Collect the supernatant, filter through a microporous membrane, and store as the test solution. Chromatographic conditions: Column: C_18_ column (5 μm particle size, 150 mm × 3.9 mm) or equivalent. Mobile Phase: Methanol + Water = 24 + 76 (*v*/*v*). Flow Rate: 1.0 mL/min. Detection Wavelength: 272 nm. Column Temperature: 25 °C. Injection Volume: 10 μL. Inject the test solution into the high-performance liquid chromatograph (HPLC). Identify caffeine based on retention time and record the peak area. Determine the caffeine concentration in the test solution using a pre-established standard curve. Perform no less than two parallel measurements to ensure accuracy. Measurements were repeated three times for each data indicator.

### 2.5. Diacetyl Determination

According to the Chinese national standard for diacetyl determination [16], after preheating the diacetyl distiller with steam, add 100 mL of non-deaerated beer sample (5 °C), followed by 1–2 drops of antifoaming agent for distillation. Collect 25 mL of distillate and cool to room temperature before diluting to volume with redistilled water. Transfer 10 mL aliquots of the distillate into two dry colorimetric tubes. Add 0.5 mL of o-phenylenediamine solution to the first tube (sample) while leaving the second tube untreated (blank control). Mix thoroughly and incubate both tubes in darkness for 25 min. Subsequently, add 2 mL of hydrochloric acid solution to the first tube and 2.5 mL to the second tube. Measure the absorbance at 335 nm wavelength using the blank as a reference. Conduct no fewer than two parallel experiments.

The diacetyl concentration is calculated using the formula X_5_ = A_335_ × 2.4, where X_5_: diacetyl content in the sample (mg/L), A_335_: absorbance measured at 335 nm wavelength using a 10 mm quartz cuvette, and 2.4: conversion coefficient between absorbance and diacetyl content for a 10 mm quartz cuvette.

### 2.6. Electronic Tongue and Electronic Nose Measurement

This experiment utilized the AIRSENSE PEN3 electronic nose system (Airsense Corporation, Schwerin, Germany) to analyze T28 CGB finished products. Instrument parameters were set as follows: 120 s sampling per group, 120 s sensor self-cleaning, 5 s sensor zeroing, 5 s sample preparation, 300 mL/min injection flow rate, and 120 s analysis duration. Data collected at the 120 s mark were used for summarization and analysis, with triplicate samples designated as A1, A2, and A3 [16].

Concurrent electronic tongue analysis was performed using the INSENT SA402B system (Insent Company, Tokyo, Japan). Samples were loaded into an automatic sampler, where specific taste sensors processed flavor information through a pattern recognition system before transmitting data to the processor. Testing protocol included 90 s electrode cleaning in positive/negative detection solutions, followed by sequential 120 s rinses in two reference solutions. After 30 s sensor equilibration, 30 s sample testing commenced. Sensors subsequently underwent dual 3 s reference solution rinses and 3 s aftertaste testing in a fresh reference solution. Triplicate samples were similarly labeled A1, A2, and A3.

### 2.7. HS-SPME-GC/MS Detection Conditions

Analytical instruments: Gas chromatography mass spectrometry (Model: PAL RTC 120-8890-5977B, Agilent company, Santa Clara, CA, USA) [18]. SPME conditions: Heating oven temperature: 60 °C; sample bottle equilibration time: 10 min; fiber head aging station temperature: 240 °C; fiber head aging time: 10 min; sample adsorption time: 20 min; sample desorption time: 2 min; GC cycle time: 35 min; sample bottle volume: 20 mL. Chromatographic conditions: The sample was injected into the GC-MS system for analysis in split-less injection mode, with an injection volume of 1 μL. After separation on a VF-WAXms capillary column (25 m × 0.25 mm × 0.2 μm, Agilent CP9204), the sample entered the mass spectrometer for detection. The injection port temperature was maintained at 240 °C, and the carrier gas was high-purity helium with a flow rate of 1.5 mL/min. The septum purge flow rate was 3 mL/min. The temperature program was as follows: initial temperature at 40 °C, equilibration for 2 min, then increasing at a rate of 5 °C/min to 100 °C, then increasing at a rate of 15 °C/min to 230 °C and maintaining for 5 min, followed by a 2-min run at 230 °C. Mass spectrometry conditions: Electron impact ionization source(EI), with a transmission line temperature of 280 °C, ion source temperature of 230 °C, quadrupole rod temperature of 150℃, and electron energy of 70eV. The scanning mode is full scan mode (SCAN), with a mass scanning range of *m/z* 50–500 and a scanning frequency of 3.2 scans per second.

To evaluate the stability of the analysis system during the computer operation, three quality control samples (Quality Control, QC) were prepared during the experimental process. The QC sample was made by mixing all the test samples and processed in the same way as the formal sample. During the instrument detection process, three QC samples were dispersed and inserted. In data analysis, the repeatability of the QC samples can reflect the stability of the instrument throughout the analysis process. It can also be used to discover variables with large variations in the analysis system, ensuring the reliability of the results.

### 2.8. Data Processing and Annotation

The raw data include QC samples and detection samples. In order to better analyze the data, a series of preprocessing steps are performed on the raw data, including filtering low-quality peaks, filling missing values, normalization, and evaluating the relative standard deviation of QC samples. Differential analysis on the preprocessed matrix file should be performed. This study first analyzed the overall differences between the two groups through PCA analysis and PLS-DA analysis (Version1.6.2). Then, differential metabolites were screened by analyzing the VIP values of metabolites in OPLS-DA (using the VIP values of PLS-DA if OPLS-DA overfits, Version1.6.2), and fold change and *p*-value in univariate analysis and volcano plots were plotted. The criteria for screening differential metabolites include (1) Fold Change = 1, which means that if the difference in metabolites between the control group and the experimental group is more than 1-fold, the difference is considered significant. (2) The VIP of OPLS-DA model is ≥1. (3) *p*-value < 0.05. Through this section, significant differences in metabolites between the two comparison groups can be identified. Differential metabolites were identified by screening the KEGG pathway database (https://www.kegg.jp/kegg/pathway.html, accessed on 18 December 2024) and the Human Metabolome Database (HMDB, https://hmdb.ca/, accessed on 18 December 2024). Annotating metabolic pathways in the KEGG database allowed for obtaining pathways involving differential metabolites. We performed pathway enrichment analysis using the Python software package (Scipy, Version1.0.0) and obtained the most relevant biological pathways for experimental treatment through Fisher’s exact test. Origin 2022 software was used to create heat maps and other images.

## 3. Results

### 3.1. Changes in Physical and Chemical Indicators During the Fermentation Process of Coffee-Grounds Craft Beer

#### 3.1.1. Intelligent Sensory Evaluation

Electronic nose and electronic tongue technology have important applications in food flavor analysis, which can quickly and accurately detect volatile components and basic tastes in food [17]. According to the response of the electronic tongue sensor, CGB has the characteristics of bitterness, umami, and salty taste (Figure 2A), which are mainly caused by different chemical components such as caffeine, amino acids, and minerals [19]. The response of coffee-grounds craft beer to bitterness sensors is relatively high, indicating that it contains high bitterness components such as caffeine and theophylline [20]. During the fermentation process, we detected the presence of a bitter-tasting metabolite, ethyl phenylacetate, which was consistently identified in our analysis. The emergence of this compound contributes significantly to the bitter flavor profile observed in CGB.

Figure 2B shows the response radar diagram of the CGB electronic nose sensor. The sensing values of coffee-grounds brewed beer to W5S, W1S, W1W, W2S, and W2W are 103.897, 47.540, 120.828, 84.228, and 68.759, respectively. The higher values indicate that CGB contains higher contents of nitrogen oxides, methyl compounds, inorganic sulfides, alcohols, aldehydes and ketones, and organic sulfides. The sensing values of W1C, W3C, W6S, W5C, and W3S are 0.177, 0.459, 2.952, 0.736, and 1.639, respectively. The low values indicate that CGB has a low content of aromatic compounds, ammonia compounds, hydrides, short-chain alkanes, and long-chain alkanes.

#### 3.1.2. Changes in Physical and Chemical Indicators During Fermentation Process

Figure 3A–K shows the changes in physicochemical properties, such as alcohol content and diacetyl content of CGB, during the fermentation process. During the fermentation process, the alcohol content and original wort concentration of CGB show an upward trend, which is mainly attributed to the yeast decomposing fermentable sugars into ethanol and carbon dioxide, thereby continuously accumulating ethanol [21]. In addition, the lignocellulose in coffee grounds gradually hydrolyzes in the fermentation broth, with hemicellulose generating fermentable sugars such as xylan through acidic hydrolysis, providing continuous fermentation substrates for yeast and further promoting the increase in alcohol content.

As the fermentation time prolongs, the overall diacetyl content and bitterness value of CGB show a significant downward trend. The decrease in diacetyl during the 14th to 21st day of fermentation (T14~T21) is particularly significant, which may be due to the fact that during the fermentation period (T0~T14), yeast preferentially synthesizes α—acetolactate to meet the demand for valine biosynthesis, while in the post ripening stage (T14~T28), yeast gradually enters a metabolic equilibrium state and inhibits α—acetolactate synthesis through the regulation mechanism of thiamine pyrophosphate (TPP)—dependent pyruvate decarboxylase (PDC). At the same time, yeast in the post-ripening stage (T14~T28) activates the secondary metabolic pathway, activates NADH-dependent diacetyl reductase in mitochondria, and gradually reduces diacetyl to ethylene glycol and 2,3-butanediol. As the flavor threshold of ethylene glycol and 2,3-butanediol is 10 times and 100 times higher than that of diacetyl, this conversion process significantly reduces the adverse flavor effects of diacetyl in beer, thereby improving the overall flavor quality of CGB.

The bitterness value showed a significant decrease from the 7th to the 21st day of fermentation (T7~T21), while it significantly increased on the 28th day (T28). The change in bitterness value is closely related to the degree of beer fermentation, which is mainly determined by the inherent bitterness substances in hops and coffee. The iso-alpha acids produced during the boiling stage (the source of hop bitterness) are relatively stable in the early stages of fermentation, resulting in a higher bitterness value. However, as fermentation progresses, bitter substances may precipitate with yeast, proteins, etc., causing some bitter substances to undergo hydrolysis or structural changes. Although the generation of flavor compounds such as alcohol and esters can, to some extent, mask bitterness, the actual decrease in concentration of bitter substances is limited. Finally, in the late stage of fermentation (T28), due to the rerelease of bitter substances or the accumulation of other metabolites, the bitterness value shows an upward trend again.

With the extension of fermentation time, the total acid and pH values of CGB did not show significant changes, only showing a slightly increasing trend [22], while the overall content of chlorogenic acid showed an upward trend. This phenomenon may be attributed to the formation of pH buffering pairs between polyphenolic substances such as chlorogenic acid and quinic acid in coffee grounds and organic acids such as lactic acid and acetic acid produced during fermentation, thereby controlling pH fluctuations within a certain range. Meanwhile, coffee oil wraps organic acid molecules through hydrophobic interactions, reducing their effective concentration in the liquid phase, while coffee cellulose adsorbs organic acids through hydrogen bonds, forming a dynamic slow-release system and further stabilizing the pH value. The caffeine content showed an overall decreasing trend during the fermentation process but increased between the 21st and 28th days (T21–T28). This may be due to an increase in cellulase activity during the later stages of fermentation (after T14 days), which degrades cellulose in coffee grounds and releases physically encapsulated caffeine. In addition, the dissociation of lignin carbohydrate complex (LCC) exposes bound caffeine, leading to an increased dissolution rate of caffeine in the later stage. The overall color and protein content of CGB show an upward trend, which may be due to the yeast entering the decay period in the later stage of fermentation (>21 days), triggering cell autolysis and releasing a large number of intracellular proteins, including proteases, β—glucanases, etc. These enzymes further degrade cell wall polysaccharides. At the same time, the release of cytoskeletal components such as actin and microtubule proteins, as well as the release of bound proteins during the hydrolysis of coffee grounds cellulose, also contributed to the increase in protein content. The increase in the color of beer may depend on melanoid compounds, oxidation of malt components, Maillard reaction products, and excipient components [23]. Melanoids in coffee grounds gradually dissolve with cellulose degradation, and their conjugated double-bond structure directly contributes to color (Table 1). In addition, polyphenol oxidase catalyzes the oxidation of phenolic substances such as chlorogenic acid and caffeic acid to quinones, which further polymerize into melanoidins, resulting in a significant increase in chromaticity [24].

Through Pearson correlation analysis, we further investigated the relationships between various physicochemical indicators in CGB, with the correlation heatmap presented in Figure 3L. The results revealed that alcohol content showed positive correlations with original gravity, real fermentation degree, pH, chroma, chlorogenic acid, and caffeine. Notably, statistically significant differences (*p* < 0.05) were observed between alcohol content and both original gravity and pH. This phenomenon may be attributed to higher original gravity indicating elevated initial fermentable sugar content (glucose and maltose) consequently leading to increased ethanol production. Conversely, alcohol content exhibited negative correlations with diacetyl, total acid, and bitterness values.

Original gravity demonstrated positive correlations with real fermentation degree, pH, chroma, chlorogenic acid, and protein content, particularly showing a significant difference with pH (*p* < 0.05). This observation could be explained by the higher concentrations of buffering substances such as phosphates and polyphenols in high-gravity wort, which contribute to pH stability. Additionally, alkaline minerals (e.g., potassium and magnesium) present in coffee grounds might neutralize organic acids generated during fermentation. These findings collectively reflect the complex biochemical transformations occurring during CGB fermentation, which ultimately influence the texture and flavor profiles of alcoholic products.

### 3.2. Analysis of Differential Metabolites During the Fermentation Process of Coffee-Grounds Craft Beer

#### 3.2.1. Metabolite Differential Analysis Based on PCA and PLS-DA

Principal PCA is typically employed to reduce data dimensionality while retaining the majority of the original multivariate information [25]. PCA was used to analyze the overall metabolite differences and sample variability between each group of samples (Figure 4A). The results show that the QC sample grouping was good, indicating that the biological analysis and data quality were good and that the inter-group separation was due to the differential variables between groups rather than differences in the analysis process. As shown in Figure 4A, all sample coordinate points are distributed within the 95% confidence interval. The variance contribution rates of PC1 and PC2 are 36.3% and 15.7%, respectively, with a cumulative contribution rate of 52%. This indicates that these two principal components are effective in capturing the main feature information of the coffee-grounds craft beer (CGB) sample. The quality control (QC) samples are tightly clustered near the origin of the coordinates, indicating stable equipment performance and experimental conditions during data collection, which ensured the accuracy and reliability of the data. The large distance between the T7 and T14 sample groups suggests that there was not much correlation between these two groups. In contrast, the relative proximity between the T21 and T28 sample groups indicates a high degree of reproducibility of the parallel samples within each group. Therefore, it is hypothesized that the metabolic pattern of CGB changes at different fermentation stages [26,27,28,29].

Further, we investigated the differences between the four groups of samples, establishing a PLS-DA model and using supervised PLS-DA to further analyze the similarities and differences among the groups (Figure 4B). The four groups of samples with different fermentation days were clearly separated in the PLS-DA plot, and their distribution trends were similar to the PCA results. In PLS-DA analysis, R^2^_Y_ and Q^2^ are used to evaluate the explanatory and predictive abilities of the PLS-DA model, respectively [30]. We selected 200 random permutation tests to further evaluate its effectiveness.

From Figure 4C, it can be seen that the intercept between the Q^2^ regression line and the Y-axis, which is less than 0.05, indicates that the model is robust and reliable without overfitting [31]; that is, there is no overfitting, indicating that the model has good predictive ability and enhances the confidence level of the reliability of the PLS-DA model. It can verify the differences between the four groups of samples. Additionally, from the correlation heatmap (Figure 4D), it can be clearly seen that there are differences between sample groups based on color differences, which is consistent with the PCA results [32].

#### 3.2.2. Screening of Significant Differential Metabolites in the Fermentation Process of Coffee-Grounds Craft Beer

Firstly, we analyzed the overall differences between the two groups through PCA analysis and PLS-DA analysis. Then, VIP values of metabolites in OPLS-DA analysis and fold change and *p*-value in univariate analysis were used to screen for differential metabolites, and volcano plots were plotted. If the fold change is one, that is, if the difference in metabolites between the control group and the experimental group is more than one time, it is considered significant. Therefore, through OPLS-DA analysis, significant differential metabolites were screened using VIP ≥ 1 and *p*-value < 0.05 as screening criteria. A total of 183 differential metabolites were detected in CGB from 4 different fermentation days, and a total of 86 significant differential metabolites were detected from four fermentation days (Table 2).

The volcano plots (Figure 5A–F) show the differential levels of CGB metabolites at different fermentation days, with red indicating upregulation and blue indicating downregulation. The results show that there were 19 significantly different metabolites (4 upregulated and 15 downregulated) in the T7 and T14 comparison groups, 71 significantly different metabolites (25 upregulated and 46 downregulated) in the T7 and T21 comparison groups, 69 significantly different metabolites (29 upregulated and 40 downregulated) in the T7 and T28 comparison groups, 64 significantly different metabolites (32 upregulated and 32 downregulated) in the T14 and T21 comparison groups, 61 significantly different metabolites (32 upregulated and 29 downregulated) in the T14 and T28 comparison groups, and 32 significantly different metabolites (21 upregulated and 11 downregulated) in the T21 and T28 comparison groups. From this, it can be seen that the largest differences in the metabolome are between T7 and T21, with a total of 71 differential metabolites. This may be due to the time-controlled release and redox reactions of coffee grounds during the T7 and T21 periods. At T7, the free polyphenols in coffee grounds quickly dissolve, giving a sense of astringency and antioxidant activity. But at T21, the bound polyphenols are gradually released through microbial enzymatic hydrolysis and oxidized, and the flavor shifts from bitterness to complex caramel flavor. Unsaturated fatty acids in coffee oil are oxidized by oxygen and enzymes to form aldehydes and ketones, contributing to nutty or grassy flavors. Fatty acids are esterified with ethanol to produce long-chain esters, which increase the roundness of the CGB.

#### 3.2.3. Metabolite Classification with Significant Differences in Fermentation Days

A total of 86 significantly different metabolites were detected from four different fermentation days in the CGB, with different colors indicating primary classification and larger areas indicating more differential metabolites in that classification [33]. The hierarchical classification (superclass, class, and subclass) of the HMDB (human metabolomic database) is mainly based on the chemical structure, biological function, and metabolic pathway of metabolites. Its classification system aims to systematically sort out the diversity of metabolites and facilitate data retrieval and analysis by researchers. Superclass: core chemical structure or biomacromolecule category based on metabolites. The top three in terms of proportion in the HMDB Superclass (Figure 6A) are lipids and lipid-like molecules: 17 (30.91%), organic oxygen compounds: 10 (18.18%), and benzenoids: 8 (14.55%), accounting for 63.64% of the total significantly different metabolites. Class: on the basis of the superclass, it is subdivided according to more specific chemical structures or functions. The top five in the HMDB Class (Figure 6B) are fatty acyls: 14 (25.45%), organooxygen compounds: 10 (18.18%), benzene and substituted derivatives: 8 (14.55%), carboxylic acids and derivatives: 5 (9.09%), and saturated hydrocarbons: 5 (9.09%), accounting for 76.36% of the total significantly different metabolites. Subclass: on the basis of class, it is further divided according to the details of chemical structure or metabolic function. The top five categories of HMDB Subclasses (Figure 6C) are carbonyl compounds: 8 (14.55%), fatty acid esters: 6 (10.91%), alkanes: 5 (9.09%), fatty alcohols: 5 (9.09%), and carboxylic acid derivatives: 4 (7.27%).

#### 3.2.4. Cluster Analysis and Correlation Analysis of Differential Metabolites Among Multiple Groups

The clustering heatmap (Figure 7A) provides a visual representation of the clustering relationship between the relative expression levels of various metabolites during CGB fermentation. Based on VIP values, the top 50 differential metabolites of CGB in the four stages were selected for hierarchical clustering analysis. The redder the color, the higher the relative expression level of metabolites, and the bluer the color, the lower the relative expression level of metabolites [34]. According to Figure 6A and the HMDB Superclass metabolite classification information displayed, the T7 and T14 metabolites are similar, with higher relative content of metabolites from Organic acids and derivatives such as acetic acid, higher relative content of metabolites from Organic oxygen compounds such as 1-Propanol, and higher relative content of metabolites from organoheterocyclic compounds such as 4-(Furan-2-yl) butan-2-one. The metabolites of 2-methyl-Propanoic acid and 3-methyl-Butanoic acid are relatively abundant. The metabolites of T21 and T28 are similar. Organoheterocyclic compounds such as 2,4,5-trimethyl-1,3-Dioxolane have a higher content of metabolites; Organic oxygen compounds such as 2-Pentanone and 2-Butanol have a higher content of metabolites; Lipids and lipid-like molecules including Diethyl succinate, Citronellyl isobutyrate, (E,E)-1,5-dimethyl-8-(1-methylethylidene)-1,5-Cyclodecadiene, 2-ethyl-1-Hexanol, Dodecanoic acid, Ethyl Oleate, Ethyl 9-hexadecenoate, and Nonanoic acid have a higher content of metabolites; Organic acids and derivatives such as Ethyl isobutyrate, Isopropyl acetate, sec-Butyl acetate, and 1-Decanol have a higher content of metabolites; Hydrocarbons like Hexadecane have a higher content of metabolites; Benzenoids including Ethyl benzoate and 2-Propylphenol have a higher content of metabolites.

Using the Pearson correlation method, the top 10 differential metabolites were selected based on VIP values, and the correlation coefficients between significantly different metabolites were analyzed (Figure 7B). The correlation coefficient (R) between differential metabolites is between −1 and +1, with R > 0 indicating a positive correlation, indicated in red; R < 0 indicates a negative correlation, represented in blue. Among them, Dodecanoic acid, ethyl ester shows a positive correlation with Isopropyl acetate and Citronellyl isobutyrate, while Citronellyl isobutyrate is positively correlated with Isopropyl acetate. 2,4,5-Trimethyl-1,3-Dioxolane exhibits a negative correlation with Benzaldehyde. Additionally, Citronellyl isobutyrate demonstrates a negative correlation with Hexadecanoic acid, ethyl ester but maintains a positive correlation with Dodecanoic acid, ethyl ester. There is a competitive inhibition relationship between ethyl ester and other substances [35].

### 3.3. Analysis of Flavor Formation Mechanism in the Fermentation Process of Coffee-Grounds Craft Beer

During the four stages of CGB fermentation, a total of 183 differential metabolites were detected and matched with KEGG’s database to obtain information on the pathways involved in metabolites. KEGG pathway enrichment analysis was performed on the differential metabolites in the four stages of the CGB fermentation process, with a total of 35 metabolic pathways identified. Using the corrected *p*-value as a reference, the top 20 metabolites were selected for pathway enrichment analysis, as shown in Figure 8. These are distributed across three categories, namely Human Diseases, Metabolism, and Organismal Systems. In Human Diseases, it is mainly enriched in one metabolic pathway, which is alcoholic liver disease. In Metabolism, it is mainly enriched in 16 pathways, mainly in glycosaminoglycan biosynthesis heparan sulfate/heparin benzoxazinoid biosynthesis, prodigiosin biosynthesis, taurine and hypotaurine metabolism, pyruvate metabolism, glycolysis/gluconeogenesis, sulfur metabolism, phenylalanine, tyrosine and tryptophan biosynthesis, C5-Branched dibasic acid metabolism, biosynthesis of alkaloids derived from histidine and purine, zeatin biosynthesis, propanoate metabolism, carbon fixation pathways in prokaryotes, toluene degradation, butanoate metabolism, and limonene degradation. In Organic Systems, its pathways are enriched in three pathways, namely protein digestion and absorption, cholinergic synapse, and carbohydrate digestion and absorption. In addition, it also involves the synthesis and metabolism of various flavor compounds such as alcohols, acids, esters, aldehydes, etc. Therefore, key metabolic pathways are also important factors leading to flavor differences in coffee-grounds craft beer [36,37].

Higher alcohols have indirect flavor importance as they are precursors of the most flavor-active esters [38]. Appropriate levels of esters can enhance the flavor of beer. Ester compounds have characteristics such as strong permeability, low flavor threshold, and rich, fruity aroma [39]. The basic factors for ester formation are the levels of two substrates (alcohol and activated fatty acyl CoA) and the relative activity of ester synthase and ester hydrolase [40]. Through hierarchical clustering analysis, it can be seen that the content of various metabolites in different fermentation day groups has a significant impact (Figure 7A). The expression levels of differential metabolites in samples T21 and T28 were higher than those in T7 and T14, while the differential metabolites with low QCU expression levels were close to 60%, more than in T21 and T28. The relative expression levels of lipids and ester-like molecules, organic oxygen compounds, benzene ring compounds, organic heterocyclic compounds, etc., were significantly lower than in T21 and T28, indicating that T21 and T28, with increasing fermentation time, give beer more flavor. Meanwhile, metabolites with relatively low expression levels in T7 and T14 showed opposite expression levels in T21 and T28, including lipids and ester-like molecules, as well as significantly higher relative contents of organic oxygen compounds and benzene ring compounds than in T7 and T14.

The addition of coffee grounds and the length of beer fermentation time can directly affect the formation of beer flavor [18]. After saccharification, the oil left behind in coffee plays a significant role in the presentation of the flavor of the CGB during its fermentation process. The synergistic effect of different esters forms complex flavors. For example, n-hexyl acetate (apple flavor) and octyl acetate (flower flavor) can significantly enhance the aroma at low concentrations [41,42]. It can be found that during the fermentation process of T21~T28, nearly 60% of the substances have relatively high expression levels. Among these differential metabolites, there are multiple lipids and ester-like molecules involved, and the expression levels in T21~T28 are significantly higher than those in T7~T14. This indicates that oil plays a crucial role in the flavor of beer during the later stages of fermentation. The natural lipids in coffee grounds are rich in terpenes such as caryophyllene and limonene, which give beer a woody, citrus, or floral tone. These fat-soluble flavor compounds are slowly released through the emulsification of fats during the fermentation process, forming a long-lasting coffee aftertaste. The free fatty acids generated by lipid hydrolysis serve as acyl donors and are catalyzed by yeast ester synthase to bind with alcohols, producing fruity esters (such as ethyl acetate and isoamyl acetate). These lipids and ester-like molecules make important contributions to the fermentation flavor and sensory quality of coffee grounds in craft beer [43].

## 4. Conclusions

Our non-targeted metabolomic investigation of coffee-grounds craft beer (CGB) fermentation revealed distinct temporal metabolite profiles that underpin its unique sensory characteristics. From 183 differential metabolites identified across four fermentation stages, the most pronounced metabolomic divergence occurred between T7 and T21 (71 differential metabolites). Statistical filtering (VIP ≥ 1, *p* < 0.05) identified 86 significantly differential metabolites predominantly categorized as lipids and lipid-like molecules (30.91%), organic oxygen compounds (18.18%), and benzenoids (14.55%). This distribution pattern reflects complex biochemical transformations during fermentation, with lipid-related compounds contributing to mouthfeel and flavor persistence, while benzenoids impart characteristic aromatic notes derived from coffee components. KEGG pathway enrichment analysis revealed significant activation of protein digestion and absorption, glycosaminoglycan biosynthesis, and benzoxazinoid biosynthesis pathways, suggesting that spent coffee grounds introduce novel substrate components that activate alternative metabolic pathways not typically dominant in conventional beer fermentation. The clear temporal separation observed in multivariate analyses demonstrates the dynamic evolution of CGB’s metabolome throughout fermentation, offering potential quality control parameters and process optimization targets. Significant correlation patterns between specific metabolites indicate interconnected biochemical networks collectively shaping CGB’s organoleptic profile. Key limitations of this study include the absence of conventional craft beer as a comparative control and insufficient characterization of yeast strain contributions to metabolite production. Future research should integrate microbiological characterization with metabolomic profiling to establish clear relationships between microbial dynamics and flavor development in CGB, while comparative studies with conventional beer would further elucidate the distinctive metabolic features attributable to coffee grounds incorporation. In conclusion, this study provides fundamental insights into the metabolomic basis of CGB’s distinctive characteristics, identifying potential targets for process optimization and product standardization in the sustainable valorization of coffee by-products through craft brewing applications.

## Figures and Tables

**Figure 1 foods-14-01014-f001:**
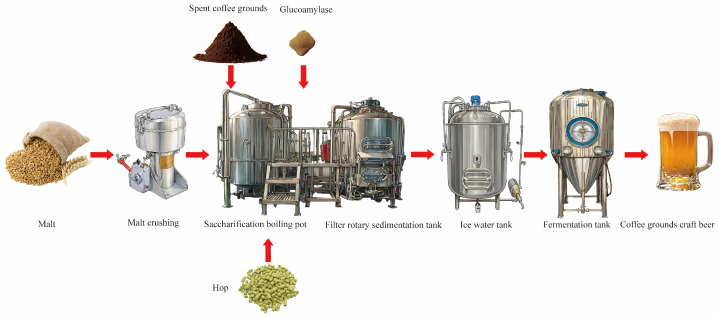
Craft beer brewing process using coffee grounds.

**Figure 2 foods-14-01014-f002:**
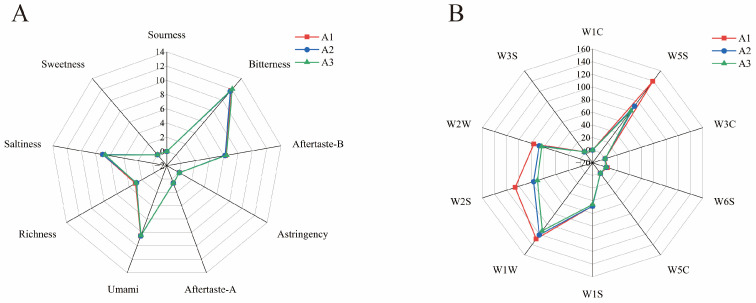
(**A**) Radar response map of electronic tongue sensor for coffee-grounds craft beer and (**B**) radar response map of electronic nose sensor for coffee-grounds craft beer.

**Figure 3 foods-14-01014-f003:**
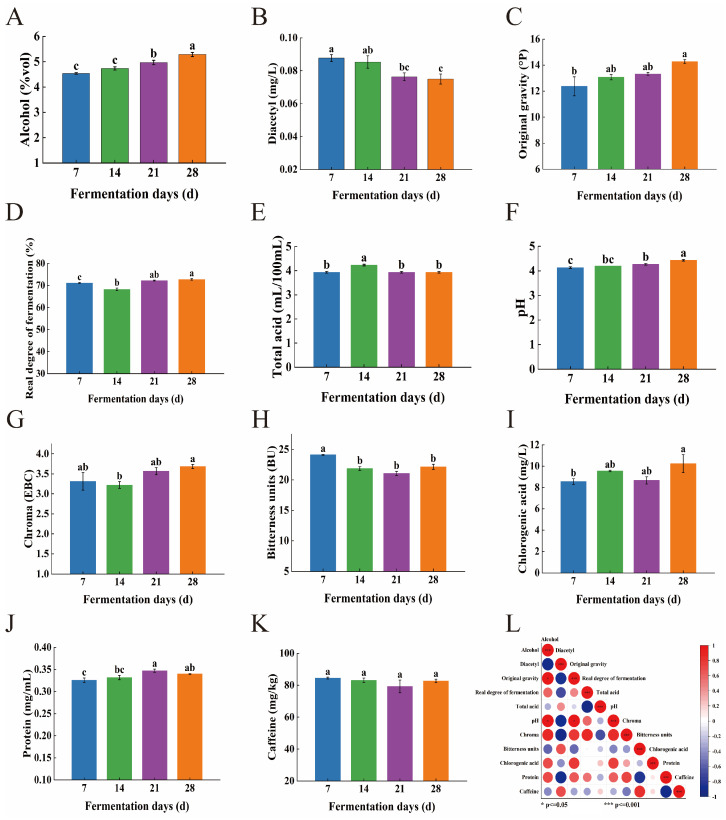
(**A**) Changes in alcohol content, (**B**) Diacetyl content, (**C**) Original gravity, (**D**) Real degree of fermentation, (**E**) Total acid, (**F**) pH, (**G**) Chroma, (**H**) Bitterness units, (**I**) Chlorogenic acid, (**J**) protein, and (**K**) caffeine, during the fermentation process of coffee-grounds craft beer. The values of different superscripts were significantly different (*p* < 0.05), while the values of the same superscripts were not significantly different (*p* > 0.05). (**L**) Pearson correlation coefficient heatmap based on basic physicochemical indicators of coffee-grounds craft beer. Positive coefficients are represented by red circles, which indicate a direct relationship between variables in the matrix, and negative coefficients are shown as blue circles, which reflect an inverse relationship. *: *p* ≤ 0.05; ***: *p* ≤ 0.001.

**Figure 4 foods-14-01014-f004:**
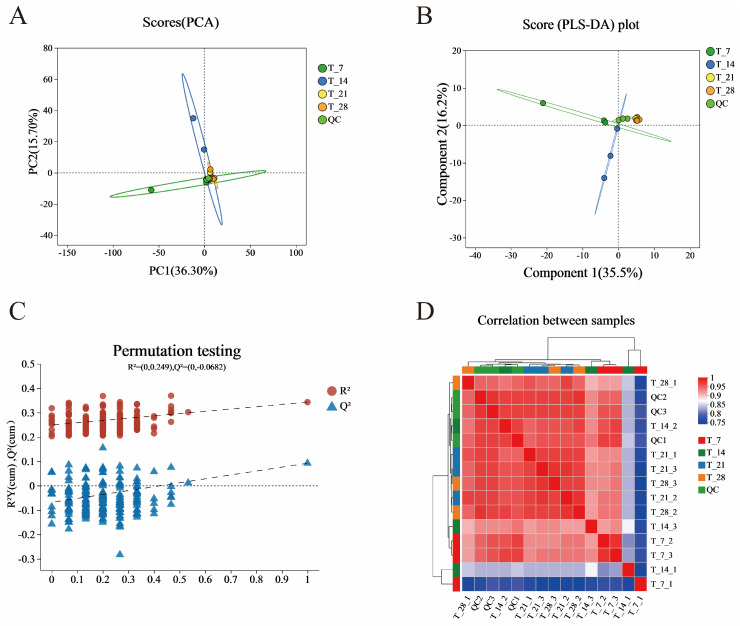
PCA analysis chart (**A**) of samples with different fermentation days, PLA-DA analysis chart (**B**) of samples with different fermentation days, PLS-DA displacement test chart (**C**) of samples with different fermentation days, and correlation heatmap of samples with different fermentation days (**D**).

**Figure 5 foods-14-01014-f005:**
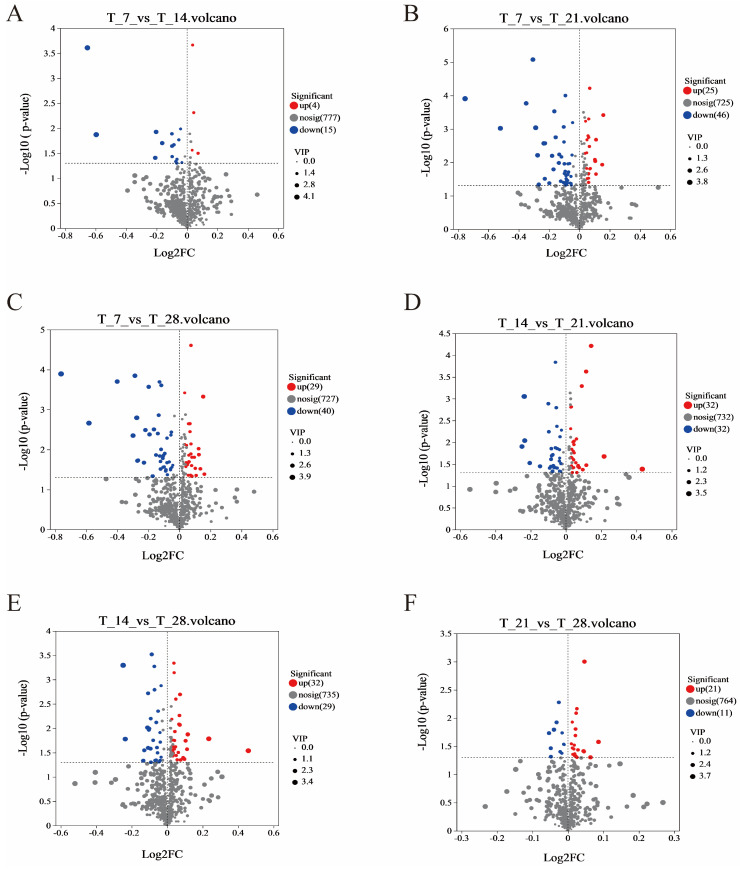
Volcanic map of significantly different metabolites in different comparison groups. (**A**) T7 vs. T14 Volcanic map; (**B**) T7 vs. T21 Volcanic map; (**C**) T7 vs. T28 Volcanic map; (**D**) T14 vs. T21 Volcanic map; (**E**) T14 vs. T28 Volcanic map; (**F**) T21 vs. T28 Volcanic map.

**Figure 6 foods-14-01014-f006:**
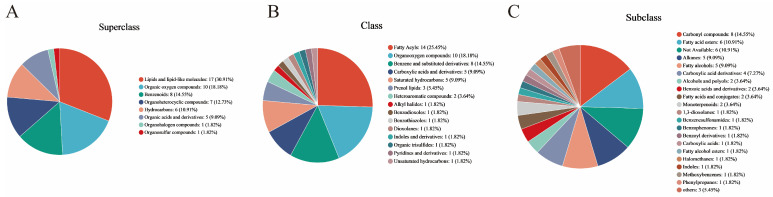
Classification of metabolites with different fermentation days: (**A**) HMDB Superclass, (**B**) HMDB Class, and (**C**) HMDB Subclass.

**Figure 7 foods-14-01014-f007:**
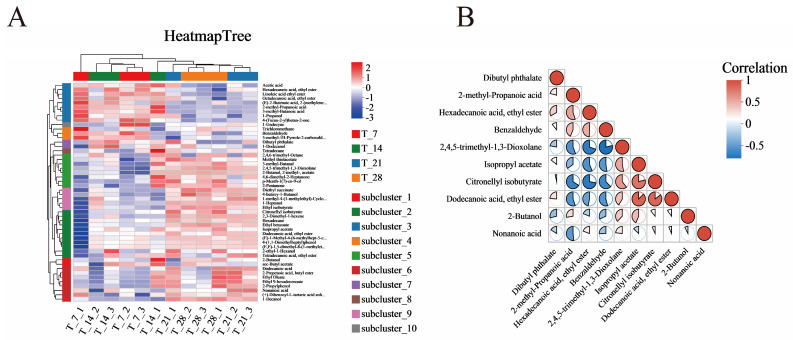
(**A**) Multi-group comparison of differential metabolite clustering heatmap. (**B**) Triangle bubble thermogram for metabolite correlation analysis. Colors denote correlation coefficients (positive/negative values indicate positive/negative correlations), while pie size reflects p-values. Absolute correlation coefficients approaching 1 signify stronger positive/negative correlations between metabolites.

**Figure 8 foods-14-01014-f008:**
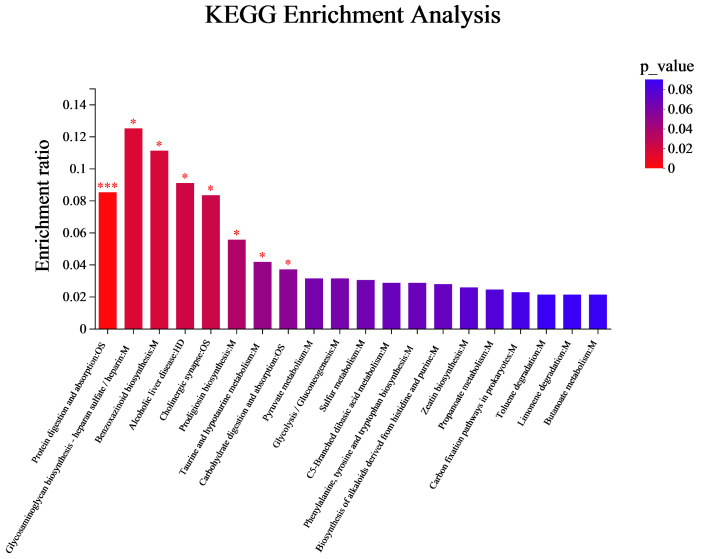
KEGG enrichment analysis of metabolites with different fermentation days. The color gradient of the columns reflects the enrichment significance of KEGG terms, where darker coloration corresponds to greater statistical significance. Enrichment levels are annotated as: *** for *p*-value < 0.001 and * for *p*-value < 0.05.

**Table 1 foods-14-01014-t001:** Chromaticity of samples with different fermentation days.

Chroma	T0	T7	T14	T21	T28
L*	21.13 ± 0.65 ^a^	17.61 ± 0.46 ^c^	17.27 ± 0.29 ^c^	19.08 ± 0.15 ^b^	19.41 ± 0.20 ^b^
a*	−1.14 ± 0.12 ^bc^	−1.27 ± 0.07 ^c^	−1.08 ± 0.04 ^b^	−0.90 ± 0.21 ^a^	−0.92 ± 0.09 ^a^
b*	0.51 ± 0.15 ^ab^	0.31 ± 0.38 ^bc^	0.08 ± 0.05 ^c^	0.64 ± 0.12 ^ab^	0.88 ± 0.11 ^a^
ΔE	3.51 ± 0.16 ^ab^	3.31 ± 0.38 ^ab^	3.22 ± 0.15 ^b^	3.57 ± 0.16 ^ab^	3.68 ± 0.10 ^a^

All data are presented as mean ± standard deviation (SD). L*: brightness, a*: redness/greenness, b*: blue/yellow. Different superscripts (a–c) within the row indicate significant differences in various color parameters under the same fermentation time.

**Table 2 foods-14-01014-t002:** Table of significantly differential metabolites in the CGB fermentation process.

Metabolite	CAS ID	Formula	Retention Time	RI	T7_Mean	T14_Mean	T21_Mean	T28_Mean
Ethyl 3-hydroxyoctanoate	7367-90-0	C_10_H_20_O_3_	13.082	1884.134	5.626	5.678	5.829	5.896
2-Furancarboxylic acid, ethyl ester	614-99-3	C_7_H_8_O_3_	10.835	1618.217	5.032	5.132	5.264	5.318
Dodecanoic acid, ethyl ester	106-33-2	C_14_H_28_O_2_	12.782	1840.466	6.708	6.95	7.114	7.024
Ethyl 9-hexadecenoate	54546-22-4	C_18_H_34_O_2_	15.215	2279.449	6.137	6.703	7.497	7.076
Tetradecanoic acid, ethyl ester	124-06-1	C_16_H_32_O_2_	14.049	2047.243	5.877	5.966	6.088	5.895
Hexadecane	544-76-3	C_16_H_34_	10.617	1598.039	6.289	6.451	6.566	6.535
Tridecane	629-50-5	C_13_H_28_	6.167	1302.205	5.695	5.595	5.892	5.828
3-Phenyl-1-propanol, acetate	122-72-5	C_11_H_14_O_2_	13.432	1940.1	4.644	4.712	4.565	4.531
2-Butanol	78-92-2	C_4_H_10_O	2.846	1042.887	8.72	9.002	8.697	9.038
4-(Furan-2-yl)butan-2-one	699-17-2	C_8_H_10_O_2_	11.071	1640.844	6.002	5.929	5.894	5.862
cis-Verbenyl angelate	00000-00-0	C_15_H_22_O_2_	11.157	1649.089	4.719	4.669	4.642	4.594
1-Decanol	112-30-1	C_10_H_22_O	12.162	1758.088	5.908	5.683	6.696	6.75
2-Tridecanone	593-08-8	C_13_H_26_O	12.531	1803.93	4.697	4.022	4.303	4.289
2-methyl-Propanoic acid	79-31-2	C_4_H_8_O_2_	10.291	1575.21	6.934	6.977	6.761	6.771
Diethyl succinate	123-25-1	C_8_H_14_O_4_	11.399	1672.291	5.908	6.2	6.482	6.618
1,5-dichloro-Pentane	628-76-2	C_5_H_10_Cl_2_	8.443	1449.805	5.457	5.621	5.737	5.711
3-Pyridinemethanamine	3731-52-0	C_6_H_8_N_2_	12.795	1842.358	4.648	4.348	4.616	4.664
Ethyl 2-methylbutyrate	7452-79-1	C_7_H_14_O_2_	2.863	1044.665	5.089	5.196	5.472	5.582
3-methyl-Butanoic acid	503-74-2	C_5_H_10_O_2_	11.412	1673.538	6.496	6.509	6.365	6.379
Acetic acid	64-19-7	C_2_H_4_O_2_	8.554	1457.004	6.749	6.768	6.632	6.443
1-(2-thienyl)-Ethanone	88-15-3	C_6_H_6_OS	12.268	1771.078	5.265	4.866	4.722	4.856
2-phenoxy-Ethanol	122-99-6	C_8_H_10_O_2_	14.567	2145.833	5.741	5.798	5.448	5.514
Ethyl isobutyrate	97-62-1	C_6_H_12_O_2_	2.184	973.6402	5.716	5.874	6.047	6.156
2-Butenoic acid, butyl ester	7299-91-4	C_8_H_14_O_2_	6.475	1322.179	4.911	4.755	4.426	4.507
Indole	120-72-9	C_8_H_7_N	16.057	2458.149	5.742	5.599	5.479	5.454
Ethyl phenylacetate	101-97-3	C_10_H_12_O_2_	12.35	1781.127	5.299	5.337	5.656	5.751
Ethyl benzoate	93-89-0	C_9_H_10_O_2_	11.29	1661.841	5.441	5.771	5.995	6.13
2-Pentanone	107-87-9	C_5_H_10_O	2.26	981.59	5.338	5.693	5.952	6.121
Piperonal	120-57-0	C_8_H_6_O_3_	15.035	2241.314	4.634	4.617	4.486	4.401
Linoleic acid ethyl ester	544-35-4	C_20_H_36_O_2_	16.421	2536.195	6.725	6.554	6.235	6.048
Camphene	79-92-5	C_10_H_16_	2.962	1055.021	4.874	4.736	4.637	4.609
1-methyl-3-(1-methylethyl)-Benzene	535-77-3	C_10_H_14_	5.433	1252.413	4.485	4.457	4.569	4.585
Benzothiazole	95-16-9	C_7_H_5_NS	13.526	1955.74	5.93	5.708	5.603	5.631
2-Propylphenol	644-35-9	C_9_H_12_O	14.56	2144.444	5.729	6.032	6.426	6.21
1,5-Dimethyl-2-pyrrolecarbonitrile	56341-36-7	C_7_H_8_N_2_	10.476	1588.165	6.392	6.322	6.345	6.338
2-Butanol, 2-methyl-, acetate	625-16-1	C_7_H_14_O_2_	2.288	984.5188	5.218	6.031	6.32	6.417
1-Dodecene	112-41-4	C_12_H_24_	5.675	1268.865	4.657	5.141	5.454	5.419
Pentanoic acid, 2-hydroxy-4-methyl-, methyl ester	40348-72-9	C_7_H_14_O_3_	9.776	1539.146	4.685	4.91	5.088	5.159
2-methyl-Quinoxaline	7251-61-8	C_9_H_8_N_2_	13.591	1966.556	6.141	6.151	6.086	6.081
4-butoxy-1-Butanol	4161-24-4	C_8_H_18_O_2_	11.409	1673.25	5.955	6.021	6.145	6.185
2-Propanoylpyrrole	1073-26-3	C_7_H_9_NO	13.964	2031.618	5.784	5.79	5.68	5.687
(S)-1-methyl-4-(1-methylethenyl)-Cyclohexene	5989-54-8	C_10_H_16_	4.248	1167.402	4.935	4.847	4.799	4.83
3-methyl-4-Hexen-2-one	72189-24-3	C_7_H_12_O	5.07	1227.736	4.507	5.077	5.576	5.496
Ethyl nicotinate	614-18-6	C_8_H_9_NO_2_	12.572	1809.898	5.233	5.263	5.346	5.39
2,4,5-trimethyl-1,3-Dioxolane	3299-32-9	C_6_H_12_O_2_	2.044	958.9958	6.313	7.069	7.429	7.635
2-(methoxymethyl)-Furan	13679-46-4	C_6_H_8_O_2_	5.881	1282.869	5.027	5.231	5.265	5.309
4-Ethyl-2-isobutyl-5-methyloxazole	4294682-24-0	C_10_H_17_NO	13.215	1903.993	4.622	4.618	4.511	4.458
3-methyl-Cyclopentene	1120-62-3	C_6_H_10_	10.617	1598.039	5.497	5.779	5.583	5.536
3-methyl-Butanal	590-86-3	C_5_H_10_O	1.902	944.1423	6.109	6.542	6.515	6.623
1,3-Dioxan-4-one, 2-heptyl-6-methyl	99902-24-6	C_12_H_22_O_3_	2.891	1047.594	3.536	4.304	4.512	4.666
Methoxyacetaldehyde diethyl acetal	4819-75-4	C_7_H_16_O_3_	3.083	1067.678	4.257	4.456	4.569	4.653
(Z)-3-methyl-4-Nonene	63830-69-3	C_10_H_20_	3.286	1088.912	5.407	5.633	5.101	5.393
2,4,4-trimethyl-Hexane	16747-30-1	C_9_H_20_	6.83	1345.201	2.756	4.336	4.648	4.672
hexahydro-1,1-dimethyl-4-methylene-4H-Cyclopenta[c]furan	344294-72-0	C_10_H_16_O	6.922	1351.167	4.607	4.923	4.944	4.869
3-(1-methylbutoxy)-2-Butanol	74810-43-8	C_9_H_20_O_2_	7.141	1365.37	3.251	4.914	4.667	4.866
2,3-Dimethyl-1-hexene	16746-86-4	C_8_H_16_	8.445	1449.935	6.026	6.235	6.456	6.433
3,3-dimethyl-Pentane	562-49-2	C_7_H_16_	8.495	1453.178	4.096	4.271	4.451	4.491
3,5-dimethyl-N-phenyl-Pyrazole-1-carboxamide	00000-00-0	C_12_H_13_N_3_O	8.921	1480.804	4.723	4.594	4.616	4.536
Benzaldehyde	100-52-7	C_7_H_6_O	9.451	1516.387	7.451	7.156	6.934	7.05
p-Amino-L-phenylalanine	943-80-6	C_9_H_12_N_2_O_2_	11.538	1685.618	4.502	4.414	4.396	4.415
Valeric acid, 4-cyanophenyl ester	00000-00-0	C_12_H_13_NO_2_	12.131	1754.289	4.817	4.767	4.818	4.861
cis-1,3-Bis(aminomethyl)cyclohexane	10340-00-8	C_8_H_18_N_2_	12.336	1779.412	4.452	4.275	4.269	4.311
3-(Hydroxyimino)-6-methylindolin-2-one	107976-73-8	C_9_H_8_N_2_O_2_	12.427	1790.564	4.989	4.84	4.797	4.76
3,4-di[1-butenyl]-Tetrahydrofuran-2-ol	00000-00-0	C_12_H_20_O_2_	12.513	1801.31	5.505	5.486	5.606	5.725
Methyl butyrylprolinate	00000-00-0	C_10_H_17_NO_3_	12.829	1847.307	4.513	4.293	4.366	4.373
1,3,3-Trimethyl-2-hydroxymethyl-3,3-dimethyl-4-(3-methylbut-2-enyl)-cyclohexene	00000-00-0	C_15_H_26_O	13.265	1912.313	5.573	5.031	5.918	5.926
1H-Indene-1-methanol, alpha-methyl-, acetate	63839-85-0	C_13_H_14_O_2_	13.285	1915.641	5.546	5.807	5.301	5.51
Triallylhydrazine	1571-11-5	C_9_H_16_N_2_	13.574	1963.727	4.99	4.939	4.821	4.875
Methylenecyclopropane	6142-73-0	C_4_H_6_	13.613	1970.216	5.254	5.276	5.132	5.157
2-Hydroxy-1,7-naphthyridine	54920-82-0	C_8_H_6_N_2_O	13.754	1993.677	5.709	5.716	5.637	5.637
Mefruside	7195-27-9	C_13_H_19_ClN_2_O_5_S_2_	13.789	1999.501	4.918	4.965	4.807	4.8
Bicyclo[2.2.2]oct-5-ene-2,3-dicarbonitrile	62249-52-9	C_10_H_10_N_2_	13.908	2021.324	4.399	4.376	4.26	4.281
Tetraethylene glycol, diacetate	22790-12-1	C_12_H_22_O_7_	13.998	2037.868	4.612	4.618	4.424	4.422
2-Methyl-1-(2-methyl-[1,3]dioxolan-2-yl)but-3-yn-2-ol	00000-00-0	C_9_H_14_O_3_	14.005	2039.154	4.892	4.95	4.869	4.893
N-(3-Acetylphenyl)-N-methylacetamide	325715-13-7	C_11_H_13_NO_2_	14.363	2105.357	6.197	6.198	6.124	6.12
5-(4-nitrophenoxymethyl)-Furane-2-carboxaldehyde	00000-00-0	C_12_H_9_NO_5_	14.378	2108.333	5.157	5.152	5.057	5.072
N-benzoyl-2-cyano-Histamine	74419-68-4	C_13_H_12_N_4_O	14.541	2140.675	5.133	5.132	5.038	5.052
(E)-2-Butenoic acid, 2-(methylenecyclopropyl)prop-2-yl ester	00000-00-0	C_11_H_16_O_2_	14.56	2144.444	5.976	6.061	5.598	5.694
alpha-(2,2-dimethylpropyl)-Benzenemethanol	62338-03-8	C_12_H_18_O	14.754	2182.937	4.923	4.923	4.86	4.865
Ehtyl 2-piperonyl carbazate	31203-56-2	C_11_H_14_N_2_O_4_	14.9	2212.712	4.26	4.388	4.312	4.336
2-(4-methyl-5-cis-phenyl-1,3-oxazolidin-2-yl)-Pyrrole	00000-00-0	C_14_H_16_N_2_O	14.954	2224.153	4.64	4.72	4.529	4.507
2-Amino-4-isopropyl-5-oxo-5,6,7,8-tetrahydro-4H-chromene-3-carbonitrile	302785-66-6	C_13_H_16_N_2_O_2_	15.19	2274.153	5.09	5.083	5.036	5.007
4-Piperidinecarboxylic acid, 1-[(4-methylphenyl)sulfonyl]-, ethyl ester	297180-07-5	C_15_H_21_NO_4_S	15.435	2326.974	4.772	4.613	4.577	4.535
6-(2-Hydroxypropan-2-yl)-4,8a-dimethyl-2,3,4,6,7,8-hexahydro-1H-naphthalen-1-ol, 1-acetate	00000-00-0	C_17_H_28_O_3_	15.58	2358.772	4.998	4.958	4.893	4.89
4-Methyl-1-(2-thienyl)-1,3-pentanedione	30984-27-1	C_10_H_12_O_2_S	15.988	2444.266	5.579	5.498	5.425	5.343
2-Acetyl-4-(1,2-dihydroxypropyl)phenyl 1,3-benzodioxole-5-carboxylate, 2TFA	00000-00-0	C_23_H_16_F_6_O_9_	15.886	2423.742	4.702	4.692	4.475	4.453

Metabolite: name of the identified metabolite; CAS ID: CAS registry number of the compound; Formula: chemical formula of the metabolite; Retention time (RT): chromatographic retention time of the compound; RI: experimentally determined retention index; Mean: mean relative abundance of the metabolite in different groups.

## Data Availability

The original contributions presented in the study are included in the article; further inquiries can be directed to the corresponding author.

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
