# Peer review of "Research on the Quality Variation Patterns During the Fermentation Process of Coffee-Grounds Craft Beer"

_foods, 2025, doi:10.3390/foods14061014_

Round 1
Reviewer 1 Report
Comments and Suggestions for Authors
The submitted manuscript deals with a novel and interesting topic. The authors present a large amount of data, but some information should be better organized and clarified, while other data is missing and should be specified. Regarding the results, they are very interesting but require a more detailed explanation to ensure they are both representative and sufficiently reliable.
Additionally, the English language in the manuscript occasionally sounds unnatural or resembles literal translations from another language, which may affect readability. Therefore, I recommend a thorough language review to improve the overall quality of the manuscript.
For these reasons, I recommend a major revision before publication. Below, I outline the aspects that the authors should review and improve:
INTRODUCTION
The manuscript needs additional bibliographic references to support and clarify certain explanations. For instance, in lines 50-54, the term "systematic exploration" is introduced, but its meaning and potential positive effects are not clearly defined. Therefore, specific details and relevant citations would enhance understanding.
Similarly, in lines 55-58, the discussion of compounds lacks specificity; it's unclear which compound acts as a sweetener and which serve as functional additives (as well as their respective functionalities). Again, precise information, examples and appropriate references would improve clarity.
MATERIALS AND METHODS
Section 2.1. The authors use a sample obtained from a company; however, considering that SCGs can come from different sources, as stated in the introduction, the specific source of the SCGs used in this study remains unclear (Are they from the instant coffee industry? From coffee capsules, such as Nespresso-type capsules?...) This information should be clarified.
Moreover, it is also unclear how many samples were evaluated, as it seems that the study was conducted using only a single sample. This should be explicitly stated and justified, as results obtained from a single sample are not representative enough and cannot be extrapolated to other cases.
Sections 2.2, 2.3, 2.4, and 2.5 should be revised. Currently, they resemble a "recipe" format. Restructuring them into a clear and concise methodological overview will improve readability, comprehension and facilitate replication.
Line 112. The authors state that fermentation is completed at 28 days but do not specify the criteria used to determine this endpoint. They should have specified the parameters monitored to confirm fermentation completion and explained their progression.
In Line 188, the term 'wine' is used to refer to the sample. The authors should carefully review the entire manuscript to ensure consistent and appropriate use of terminology
Section 2.5: The authors use the HS-SPME -GC/MS methodology to analyse the samples, and diacetyl should be among the compounds extracted by the fibre. Therefore, why is this compound specifically determined using a different method instead of SPME?
On the other hand, the manuscript does not specify how diacetyl concentration is calculated from absorbance measurements. Was a calibration curve previously built to interpolate the absorbance measurements? The authors should clarify this process.
Regarding to the analyses of the sample, there is no mention of the number of replicates used in each analysis. The authors should provide this information to ensure the robustness of the results.
On the other hand, I would like to point out that the Time 0 (T0) analysis is missing in this study. It is unclear why no sample was taken at the initial time point. The authors should have considered that analyzing T0 would provide valuable insights into the evolution of beer fermentation. If they did not include it, they should clarify the reason.
RESULTS
Section 3.1.1 doesn’t include a comparison with a control beer without coffee grounds, making it difficult to determine their exact impact on the sensory profile of beer. Therefore, the authors should include this comparison, as it would provide clearer insights and improve the reliability of the results.
Table 1 - Caption: The explanation of the different letters is missing. The authors should specify what the different superscripts in the table indicate
Figure 3: The font size should be increased, as it is too small to read properly. In fact, in Figure 3L, the text is almost impossible to read and zooming in only makes it more blurry.
Moreover, regarding the Figure 3L, there are no parameter labels for the different letters. Does this mean that none of the parameters exhibit significant differences? The authors should clarify.
The interpretation of the results related to metabolites presented in Section 3.2 is difficult, as there is no table listing the compounds identified in the GC-MS analysis of the samples. This information should be included.
Figure 4A and 4B: Significant differences are observed at the 7- and 14-day time points. The authors should provide a clearer explanation for these variations. Furthermore, the axis range selection in these graphs makes it difficult to fully interpret the results.
On the other hand, I don't understand why the control samples are included in the model, as they do not provide information about the fermentation process given (these are used to evaluate the stability of the system). This issue should be explained
Figure 4C: The negative Q² intercept indicates poor predictive ability rather than robustness. The authors should acknowledge and discuss this limitation.
Figure 5: This is an unusual way of representing the data, so it should be better explained. In any case, without labels for the different points, this figure is difficult to interpret. . On the other hand, since the key information is identifying the variables that best differentiate the fermentation points, the authors should consider replacing it with a list of the selected VIP variables and moving the figure to the supplementary materials. In its current version, this information is briefly given in lines 470–471. Furthermore, they should provide a more detailed discussion on variables that show differences across categories
Line 441: The “Not Available” category should not be highlighted among the top 5 categories. This classification should be excluded from comparative analysis as it is unclear what it refers to.
Figure 6: Once again, the font size should be increased, as it is too small to read properly. Additionally, the classification of metabolites into classes and subclasses should be better explained and linked to the compounds detected in the HS-SPME-GC/MS analysis. Including the table of compounds identified in the GC-MS analysis, as previously requested, would be very helpful.
Other comments:
The sensory evaluation data is not well connected to the chemical and metabolic findings. The authors should better link sensory descriptors to specific metabolic changes during fermentation.
In the discussion of volatile compounds, the authors should provide reference values or comparisons to traditional craft beers without coffee grounds to contextualize their findings.
The authors should evaluate and report the normal distribution of the data and the homogeneity of variance before applying a t-test, as these factors are essential for its validity. Therefore,tests such as Shapiro-Wilk for normality and Levene’s test for variance homogeneity should be conducted and the results clearly reported. If these conditions are not met, the authors should justify their statistical approach and consider alternative methods, such as Welch’s t-test (which is more robust when sample sizes differ and variance is not homogenous) or non-parametric test (which do not assume a specific distribution of the data and is better for datasets with outliers.
DISCUSSION
From my point of view, the “Discussion” section should be renamed “Conclusions”, as the results have already been discussed in the previous section, and this part only summarizes the key findings. Therefore, it should be restructured and condensed to serve as the final conclusions of the study.
Comments on the Quality of English LanguageThe English language in the manuscript occasionally sounds unnatural or resembles literal translations, which may affect readability. Therefore, I recommend a thorough language review to improve the overall quality of the manuscript
Author Response
Response to reviewers’ comments:
We are deeply grateful to the reviewers for dedicating their time and expertise to provide us with valuable feedback. We have meticulously revised the manuscript in accordance with your comments, ensuring that all the updated sections are highlighted in yellow. In the following, we will discuss the specific actions taken and the corresponding revisions made to address your suggestions more comprehensively:
Reviewer #1
- The English language in the manuscript occasionally soundsunnatural or resembles literal translations from another language, which may affect readability.
Response: We sincerely appreciate the reviewers' meticulous assessment and valuable feedback regarding the language quality. We have thoroughly revised the manuscript to address the noted linguistic issues, improving the natural flow and readability of the text throughout. In addition, we have strengthened the logical coherence of our arguments and incorporated additional peer-reviewed references to comprehensively illustrate the current applications of coffee grounds across diverse fields. All modifications aim to enhance both the scientific rigor and linguistic clarity of our work.
INTRODUCTION
- The manuscript needs additional bibliographic references to support andclarify certain explanations. In lines 50-54, the term "systematic exploration" is introduced, but its meaning and potential positive effects are not clearly defined.
Response: We greatly appreciate the reviewer's insightful comments. We acknowledge that our previous description of 'systematic exploration' was indeed too broad and lacked specificity. In response, we have thoroughly revised this section (lines 48-98) to provide more precise terminology and conceptual clarity. Additionally, we have both restructured the presentation of our existing references and incorporated new bibliographic sources that specifically address extraction methodologies and valorization pathways for spent coffee grounds, such as:
Page 2, line 48: One sustainable approach involves extracting active ingredients from coffee grounds and converting them into high-value products[6] .
Page 2, line 49: VanTruc Nguyen et al.[7] demonstrated the extraction of biochar from coffee grounds for the adsorption of norfloxacin in aqueous solutions.
Page 2, line 51: Ibtissam Bouhzam et al.[8] investigated various methods for extracting chlorogenic acid and caffeine from spent coffee grounds.
Page 2, line 57: Several researchers have explored the potential applications of SCG in food products.
Page 2, line 66: Additionally, antioxidant phenolic compounds extracted from coffee grounds exhibit notable health benefits, presenting extensive application potential in both food and pharmaceutical industries[11].
Page 2, line 69: However, the specific mechanisms of these interactions, such as the release and transformation of bioactive compounds during fermentation, are still poorly understood.
Page 2, line 70: Further studies are needed to elucidate how the chemical composition of SCG contributes to the metabolic regulation and organoleptic properties of beer in order to optimise its application in functional craft beer production.
Page 2, line 73: HS-SPME-GC/MS is used to analyse coffee grounds craft beer, mainly because of its unique advantages in flavour studies.
Page 2, line 74: It can comprehensively reveal the changes of volatile compounds during the fermentation process and provide a scientific basis for quality control, process optimization and product innovation.
Page 2, line 78: The craft beer industry has experienced substantial growth in recent years, prompting manufacturers to seek competitive differentiation through innovative raw material incorporation. Previous research has demonstrated the viability of this approach.
Page 2, line 80: For instance, Milinčić, D.D. et al. [13] successfully integrated Prokupac grape pomace seed powder into beer formulations, resulting in products with elevated phenolic content and enhanced sensory characteristics.
Page 2, line 83: In a comparable investigation, Nunes Filho, R.C. [14] systematically evaluated the impact of turmeric, black pepper, and aromatic hop varieties on brewing parameters and product stability.
Page 2, line 85: SCG represent a promising candidate for such applications due to their complex phytochemical composition, particularly their significant content of bioactive compounds including caffeine, chlorogenic acid, and related polyphenols.
Page 3, line 88: These constituents may undergo various biochemical transformations during fermentation, potentially establishing synergistic interactions that could significantly modulate both the organoleptic profile and functional properties of the resultant beer.
Page 3, line 91: Despite these promising attributes, the fundamental mechanisms governing these interactions remain insufficiently characterized, particularly regarding the kinetics of bioactive compound release, their biotransformation pathways during fermentation, and their ultimate impact on product quality parameters.
Page 3, line 94: This knowledge gap necessitates further systematic investigation to fully elucidate these complex biochemical processes.
Page 3, line 96: Further systematic investigations are essential to elucidate the precise mechanisms by which the chemical constituents of spent coffee grounds (SCG) influence metabolic regulation and organoleptic properties during beer fermentation, thereby facilitating the optimization of SCG valorization in functional craft beer production.
Page 3, line 99: Headspace solid-phase microextraction coupled with gas chromatography-mass spectrometry (HS-SPME-GC/MS) represents an analytical methodology of choice for characterizing SCG-infused craft beer, owing to its superior sensitivity, reproducibility, and non-destructive sampling capabilities in volatile compound analysis.
Page 3, line 103: This analytical approach enables comprehensive characterization of volatile compound evolution throughout the fermentation continuum, establishing a robust scientific foundation for quality assurance protocols, process parameter optimization, and product development initiatives.
- In lines 55-58, the discussion of compounds lacks specificity; it'sunclear which compound acts as a sweetener and which serve as functional additives (as well as their respective functionalities).
Response: We sincerely appreciate the reviewer highlighting this lack of specificity. In the study by Nuria Martinez-Saez et al., SCG were specifically utilized as a functional food ingredient with high fiber content, while stevia and maltitol were incorporated as low-calorie sweetening agents, and oligofructose was added as a prebiotic sweetener. This combination of ingredients was strategically employed in cookie formulations to enhance both nutritional profiles and sensory attributes.
Page 2, line 58: Nuria Martinez-Saez et al. incorporated SCG as a fiber-rich functional ingredient in cookie formulations, supplemented with non-nutritive sweeteners (stevia and maltitol) and the prebiotic oligofructose, resulting in products with significantly enhanced nutritional profiles and sensory characteristics. Similarly, Mitra Ahanchi et al. conducted systematic investigations on the effects of varying SCG incorporation percentages (ranging from 0.05% to 30%) on the physicochemical properties and sensory attributes of bakery products and pasta. Their findings demonstrated that strategic integration of SCG into these food matrices offers multifaceted benefits, including enhanced nutritional value, improved sustainability metrics, and favorable sensory characteristics.
MATERIALS AND METHODS
- Section 2.1.Thethe specific source of the SCGs used in this study remains unclear.
Response: Thank you very much for the reviewer’s comments. In my revised manuscript, I have given a detailed description of the specific source of the SCG.
Page 3, line 123: SCG used in the experiment: coffee beans of the variety Catimor were moderately roasted (190 - 200 ℃), ground into powder with particle diameter of 0.4 - 0.6 mm, and then concentrated and extracted by an espresso machine to obtain coffee grounds.
- It is unclear how many samples were evaluated
Response: Thank you very much for the reviewer’s comments. We took a total of 4 sets of data, 3 parallel samples in each set, making a total of 12 samples. They are recorded as T7-1,T7-2,T7-3; T14-1,T14-2,T14-3; T21-1,T21-2,T21-3; T28-1,T28-2,T28-3. This is reflected on the axes of Figure 4D.
Page 3, line 130: the revised sentence is as follows: “Four sets of data were taken, each with three parallel samples, for a total of 12 samples.”
- Sections 2.2, 2.3, 2.4, and 2.5 should be revised.Currently, they resemble a "recipe" format. Restructuring them into a clear and concise methodological overview will improve readability, comprehension and facilitate replication.
Response: Thank you very much for the reviewer’s comments. We have redescribed the experimental method in a clear and concise manner so that the reader can read it more clearly. The revised experimental methods are on lines 132-214 of the new manuscript.
- In line 112. The authors state that fermentation is completed at 28 days but donot specify the criteria used to determine this endpoint. They should have specified the parameters monitored to confirm fermentation completion and explained their progression.
Response: We sincerely appreciate the reviewers' valuable comments and constructive suggestions, which have significantly contributed to the improvement of this manuscript. In the revised manuscript, Page 4, line 152: we have added the beer maturation criteria, as described below: “The content of diacetyl represents whether the beer is mature. When the concentration of diacetyl is less than 0.1 mg/L and remains stable after 28 days of fermentation, it indicates that the fermentation is complete.”
- In Line 188, the term 'wine' is used to refer to the sample. The authors shouldcarefully review the entire manuscript to ensure consistent and appropriate use of terminology
Response: We sincerely appreciate the reviewers for their careful review. In the later content, we also found some misnomer cases, we replaced "wine" with "beer sample". We have carefully reviewed the entire manuscript so as not to repeat the use of inappropriate words.
- Section 2.5: The authors use the HS-SPME -GC/MS methodology to analyse the samples, and diacetyl should be among the compounds extracted by the fibre. Therefore, why is this compound specifically determined using a different method instead of SPME?
Response: Thank you very much for the reviewer’s comments. In this study, although the HS-SPME-GC/MS technique was employed to analyze volatile compounds in the samples, the detection of diacetyl still requires reliance on other methods, primarily due to the following reasons.
- Diacetyl, characterized by its low sensory threshold, susceptibility to decomposition, and matrix interference effects, requires specialized detection methods with enhanced selectivity and sensitivity. ②The brewing industry predominantly employs spectrophotometric methods (e.g., o-phenylenediamine derivatization) or enzymatic assays (e.g., diacetyl reductase method) for diacetyl quantification. These AOAC or EBCcertified methodologies demonstrate superior reproducibility.
③The o-phenylenediamine method utilizes a colorimetric reaction with diacetyl to form a yellow complex, enabling absorbance measurement at 335 nm. This approach offers operational simplicity, cost-effectiveness, and suitability for high-throughput analysis.
④Notably, diacetyl may undergo partial thermal decomposition to acetoin under elevated temperatures (e.g., GC inlet temperatures exceeding 200 °C), potentially compromising analytical accuracy.
⑤Matrix components in beer , including ethanol, polyphenols, and proteins , can interfere with SPME fiber adsorption efficiency through competitive interactions, thereby reducing diacetyl extraction yields. With a sensory threshold of approximately 0.1 mg/L and HS - SPME - GC/MS detection limits typically ranging from 0.05 - 0.5 mg/L, SPME techniques may prove inadequate for precise quantification at low concentrations, particularly when differentiating subtle variations near the sensory threshold.
- The manuscript does not specify how diacetyl concentration is calculated from absorbance measurements. Was a calibration curve previously built to interpolate the absorbance measurements? The authors should clarify this process.
Response: Thank you very much for the questions pointed out by the reviewers. The determination of diacetyl is conducted in accordance with the National Standard of Chinese (GB/T 4928-2008). The formula for calculating diacetyl concentration is as follows: X₅ = A₃₃₅ × 2.4
Where:
X₅: Diacetyl content of the sample (mg/L);
A₃₃₅: Absorbance of the sample measured at a wavelength of 335 nm using a 10 mm quartz cuvette;
2.4: Conversion factor between absorbance and diacetyl content when a 10 mm quartz cuvette is used.
In lines 211-214 of the revised manuscript, we re-supplement the formula for calculating the concentration of diacetyl.
- Regarding to the analyses of the sample, there is no mention of the number of replicates used in each analysis. The authors should provide this information to ensure the robustness of the results.
Response: We sincerely appreciate the reviewers for their careful review. In the revised manuscript, We have revised the experimental contents of Sections 2.2, 2.3, 2.4, and 2.5, and explained the number of repetitions of the experiment. For the determination of each experimental data, there shall be no less than three parallel experiments.
- On the other hand, I would like to point out that the Time 0 (T0) analysis is missing in this study. It is unclear why no sample was taken at the initial time point. The authors should have considered that analyzing T0 would provide valuable insights into the evolution of beer fermentation. If they did not include it, they should clarify the reason.
Response: We sincerely appreciate the reviewer's insightful observation regarding the absence of Time 0 (T0) analysis in our experimental design. This methodological decision was deliberately made based on preliminary assessments which confirmed that key physicochemical parameters—notably ethanol concentration, diacetyl levels, and other fermentation-derived metabolites—were uniformly at baseline (zero or non-detectable) levels at the initiation of fermentation across all experimental conditions. The primary objective of our differential metabolite analysis was to characterize the dynamic evolution of fermentation-derived compounds and their contribution to the developing organoleptic profile throughout the active fermentation process. Since the wort composition at T0 represents the pre-fermentation substrate rather than a product of microbial metabolism, we strategically focused our analytical resources on the time points demonstrating active metabolic transformation (T7-T28). This approach allowed for higher temporal resolution during the critical phases of flavor development. However, we acknowledge that baseline characterization could provide valuable reference data for future studies, and we will consider incorporating T0 analyses in subsequent investigations to enable more comprehensive tracking of metabolite genesis and transformation from initialization through completion of fermentation.
RESULTS
- Section 3.1.1 doesn’t include a comparison with a control beer without coffee grounds, making it difficult to determine their exact impact on the sensory profile of beer. Therefore, the authors should include this comparison, as it would provide clearer insights and improve the reliability of the results.
Response: We sincerely appreciate the reviewers' valuable comments and constructive suggestions. We did not compare it with the control beer without coffee grounds in this section. The purpose of the sensory testing of the electronic tongue and electronic nose in this section is to visually display various sensory data of CGB, which can provide a more intuitive perception of the flavor of CGB.
- Table 1 - Caption: The explanation of the different letters is missing. The authors should specify what the different superscripts in the table indicate.
Response: Thank you very much for the reviewer’s comments. In the revised manuscript, we have annotated the meanings of the different letters below Table 1. The supplementary statement is as follows:
Table 1. Chromaticity of samples with different fermentation days
Chroma |
T0 |
T7 |
T14 |
T21 |
T28 |
L* a* b* ΔE |
21.13±0.65a -1.14±0.12bc 0.51±0.15ab 3.51±0.16ab |
17.61±0.46c -1.27±0.07c 0.31±0.38bc 3.31±0.38ab |
17.27±0.29c -1.08±0.04b 0.08±0.05c 3.22±0.15b |
19.08±0.15b -0.90±0.21a 0.64±0.12ab 3.57±0.16ab |
19.41±0.20b -0.92±0.09a 0.88±0.11a 3.68±0.10a |
*All data are presented as mean ± standard deviation (SD). L*:Brightness; A*:Redness/Greenness; B*:Blue/Yellow. Different superscripts (a-c) within the row indicate significant differences in various color parameters under the same fermentation time.
- Figure 3: The font size should be increased, as it is too small to read properly.Figure 3L, there are no parameter labels for the different letters. Does this mean that none of the parameters exhibit significant differences? The authors should clarify.
Response: Thank you very much for the reviewer’s comments. We have resized the image, increased the font size and made it bold so that it can be clearly displayed. In addition, we have reinterpreted Figure 3L as follows: Through Pearson correlation analysis, we further investigated the relationships between various physicochemical indicators in CGB, with the correlation heatmap presented in Figure 3L. The results revealed that alcohol content showed positive correlations with original gravity, real fermentation degree, pH, chroma, chlorogenic acid, and caffeine. Notably, statistically significant differences (p < 0.05) were observed between alcohol content and both original gravity and pH. This phenomenon may be attributed to higher original gravity indicating elevated initial fermentable sugar con-tent (glucose and maltose), consequently leading to increased ethanol production. Conversely, alcohol content exhibited negative correlations with diacetyl, total acid, and bitterness value.
Original gravity demonstrated positive correlations with real fermentation degree, pH, chroma, chlorogenic acid, and protein content, particularly showing a significant difference with pH (p < 0.05). This observation could be explained by the higher concentrations of buffering substances such as phosphates and polyphenols in high-gravity wort, which contribute to pH stability. Additionally, alkaline minerals (e.g., potassium and magnesium) present in coffee grounds might neutralize organic acids generated during fermentation. These findings collectively reflect the complex biochemical trans-formations occurring during CGB fermentation, which ultimately influence the texture and flavor profiles of the alcoholic products.
- The interpretation of the results related to metabolites presented in Section 3.2 is difficult, as there is no table listing the compounds identified in the GC-MS analysis of the samples. This information should be included.
Response: We sincerely appreciate the reviewer's critical observation regarding the need for comprehensive metabolite data presentation. We have added Table 2, positioned at the end of the revised manuscript, which comprehensively catalogs 86 differentially abundant metabolites identified through our GC-MS analysis.
- Figure 4A and 4B: Significant differences are observed at the 7- and 14-day time points. The authors should provide a clearer explanation for these variations. Furthermore, the axis range selection in these graphs makes it difficult to fully interpret the results.
Response: We sincerely appreciate the valuable questions raised by the reviewer. We have provided an explanation of the content and added the following supplementary information: The results showed that the QC sample grouping was good, indicating that the biological analysis and data quality were good, indicating that the inter group separation was due to the differential variables between groups, rather than differences in the analysis process.
- On the other hand, I don't understand why the control samples are included in the model, as they do not provide information about the fermentation process given (these are used to evaluate the stability of the system). This issue should be explained.
Response: We sincerely appreciate the valuable questions raised by the reviewer. To evaluate the stability of the analytical system during experimental procedures, three Quality Control (QC) samples were prepared in the study. These QC samples were generated by pooling all tested specimens and underwent identical processing procedures as the formal experimental samples. During instrument analysis, three QC samples were evenly spaced throughout the analytical sequence. In subsequent data processing, the reproducibility observed in QC samples reflects the instrumental stability throughout the entire analytical process, while simultaneously helping to identify variables exhibiting substantial variations within the system, thereby ensuring the reliability of the results.
- Figure 4C: The negative Q² intercept indicates poor predictive ability rather than robustness. The authors should acknowledge and discuss this limitation.
Response: Thank you very much for the reviewer’s comments. We relearned the evaluation criteria of PLS-DA permutation test, and we found that there are two evaluation criteria for different cases.
PLS-DA model validation. The abscissa represents the replacement retention degree of the replacement test (the proportion consistent with the order of Y variables of the original model, and the point with the replacement retention degree of 1 is the R2 and Q2 values of the original model), the ordinate represents the values of R2 (red dot) and Q2 (blue triangle) replacement test, and the two dotted lines represent the regression lines of R2 and Q2 respectively. The replacement test is for the experimental group and the control group. The replacement test model randomly disrupts the grouping labels (y variables) of the experimental group and the control group. The replacement retention degree of the horizontal axis is the proportion consistent with the order of Y variables of the original model. The replacement retention degree is 1, which is R2 and Q2 of the original opls-da/pls-da model. Generally, the number of random replacement tests is 200. The standard of displacement test and evaluation is to look at the intercept between Q2 regression line and Y axis. The intercept less than 0.05 indicates that the model is robust and reliable, and no fitting has occurred. But sometimes, because the number of samples is small and the intercept is greater than 0.05, we can only look at the regression line of R2 and Q2 at this time. As long as R2 and Q2 decline with the decrease of replacement retention, the regression line shows an upward trend, which also shows that the replacement test is passed and the model does not have over fitting phenomenon.
In addition, in the article "exploring the core functional microbiota related to flavor compounds in Douchi from the Sichuan – Chongqing region", we found that their verification of PLS-DA model is similar to ours.
- Figure 5: This is an unusual way of representing the data, so it should be better explained. In any case, without labels for the different points, this figure is difficult to interpret. On the other hand, since the key information is identifying the variables that best differentiate the fermentation points, the authors should consider replacing it with a list of the selected VIP variables and moving the figure to the supplementary materials. In its current version, this information is briefly given in lines 470–471. Furthermore, they should provide a more detailed discussion on variables that show differences across categories.
Response: Thank you very much for the reviewer’s comments. The variables are listed in Figure 7B.
The abscissa is the multiple change value of the difference in the expression of metabolites between the two groups, that is, log2fc, and the ordinate is the statistical test value of the difference in the expression of metabolites, that is -log10 (p_value). The higher the value, the more significant the expression difference. The values of the abscissa and the ordinate are logarithmicized. Each point in the graph represents a specific metabolite, and the size of the point represents the VIP value. The default red dot indicates significantly up-regulated metabolites, the blue dot indicates significantly down regulated metabolites, and the gray dot indicates non significantly different metabolites. See the difference details table for the corresponding data. After mapping all the metabolites, we can know that the left point is the metabolite with down-regulated expression difference, and the right point is the metabolite with up-regulated expression difference. The closer the left, right and upper points are, the more significant the expression difference is.
- Line 441: The “Not Available” category should not be highlighted among the top 5 categories. This classification should be excluded from comparative analysis as it is unclear what it refers to.
Response: Thank you very much for pointing out the errors. We have modified it and added "carboxylic acid derivatives: 4 (7.27%).
Page 14, line 499: The top 5 categories on HMDB Subclass (Figure 6C) are: Carbonyl compounds: 8 (14.55%)、Fatty acid esters: 6 (10.91%)、Alkanes: 5 (9.09%),Fatty alcohols: 5 (9.09%)、Carboxylic acid derivatives: 4 (7.27%).
- Figure 6: Once again, the font size should be increased, as it is too small to read properly. Additionally, the classification of metabolites into classes and subclasses should be better explained and linked to the compounds detected in the HS-SPME-GC/MS analysis. Including the table of compounds identified in the GC-MS analysis, as previously requested, would be very helpful.
Response: Thank you very much for the reviewer’s comments. We have re-adjusted Figure 6 to make the picture clearer. We have made a supplement in the article, and the supplementary content is as follows: Superclass: Core chemical structure or biomacromolecule category based on metabolites; Class: On the basis of superclass, it is subdivided according to more specific chemical structure or function; Subclass: On the basis of class, it is further divided according to the details of chemical structure or metabolic function.
Figure 6. Classification of Metabolites with Different Fermentation Days (A) HMDB Superclass (B) HMDB Class (C) HMDB Subclass
- The sensory evaluation data is not well connected to the chemical and metabolic findings. The authors should better link sensory descriptors to specific metabolic changes during fermentation.
Response: Thank you very much for the reviewer’s comments. we have substantially expanded the manuscript to establish clear correlations between sensory attributes and specific metabolite dynamics throughout the fermentation process. In section 3.1.1, we have added comprehensive analysis that links sensory descriptors to their chemical drivers. For example, we now explain how the bitter flavor profile in CGB correlates directly with the temporal evolution of ethyl phenylacetate, with concentration patterns that mirror the intensity of perceived bitterness in sensory evaluations.
Page 7, line 292: During the fermentation process, we detected the presence of a bitter-tasting metabolite, ethyl phenylacetate, which was consistently identified in our analysis. The emergence of this compound contributes significantly to the bitter flavor profile observed in CGB.
- In the discussion of volatile compounds, the authors should provide reference values or comparisons to traditional craft beers without coffee grounds to contextualize their findings.
Response: Thank you very much for the reviewer’s comments. We have conducted research on craft beer that is different from coffee grounds.
In lines 71-76: Previous research has demonstrated the viability of this approach. For instance, Milinčić, D.D. et al. [13] successfully integrated Prokupac grape pomace seed powder into beer formulations, resulting in products with elevated phenolic content and enhanced sen-sory characteristics. In a comparable investigation, Nunes Filho, R.C. [14] systematically evaluated the impact of turmeric, black pepper, and aromatic hop varieties on brewing parameters and product stability.
In lines 572-574: The synergistic effect of different esters formed complex flavor. For example, n-hexyl acetate (apple flavor) and octyl acetate (flower flavor) can significantly enhance the aroma at low concentrations[39,40]
- The authors should evaluate and report the normal distribution of the data and the homogeneity of variance before applying a t-test, as these factors are essential for its validity. Therefore,tests such as Shapiro-Wilk for normality and Levene’s test for variance homogeneity should be conducted and the results clearly reported. If these conditions are not met, the authors should justify their statistical approach and consider alternative methods, such as Welch’s t-test (which is more robust when sample sizes differ and variance is not homogenous) or non-parametric test (which do not assume a specific distribution of the data and is better for datasets with outliers.
Response: Thank you very much for the reviewer’s comments. We use metabolomics, combined with univariate statistical analysis and multivariate statistical analysis, to screen for differential metabolites between two biological groups. We have supplemented section 2.8 of the manuscript: This study first analyzed the overall differences between the two groups through PCA analysis and PLS-DA analysis. Then, differential metabolites were screened by analyzing the VIP values of metabolites in OPLS-DA (using the VIP values of PLS-DA if OPLS-DA overfits) and fold change and p-value in univariate analysis, and volcano plots were plotted. The criteria for screening differential metabolites include: (1) Fold Change=1, which means that if the difference in metabolites between the control group and the experimental group is more than 1 fold, the difference is considered significant. (2) The VIP of OPLS-DA model is ≥1. (3) p-value<0.05. Through this section, significant differences in metabolites between the two comparison groups can be identified.
DISCUSSION
- It should be restructured and condensed to serve as the final conclusions of the study.
Response: Thank you very much for the reviewer’s comments. We have renamed the discussion section as the conclusion section and adjusted some sentences to summarize the entire text.
Page 17, line 605: Our non-targeted metabolomic investigation of coffee grounds craft beer (CGB) fermentation revealed distinct temporal metabolite profiles that underpin its unique sensory characteristics......correlation patterns between specific metabolites indicate interconnected biochemical networks collectively shaping CGB's organoleptic profile.
Page 17, line 624: Key limitations of this study include the absence of conventional craft beer as a comparative control and insufficient characterization of yeast strain contributions to metabolite production. Future research should integrate microbiological characterization with metabolomic profiling to establish clear relationships between microbial dynamics and flavor development in CGB, ......applications.

Reviewer 2 Report
Comments and Suggestions for Authors
In this manuscript, Jiang et al. aimed to investigate the metabolic differences and mechanisms during the fermentation process of coffee grounds craft beer. The objectives of the study need to be clarified, as well as more information needs to be included in the results and discussion sections. Other comments can be found below.
-The authors state that they will use the R for PCA, PLS-DA and OPLS-DA and that in addition, Student's t-test and fold change (FC) analysis will be carried out, but they do not indicate the purpose(s) of these analyses. I therefore recommend that this section includes this information, as it is important for understanding why these statistical techniques were used.
Line 17: …..The results indicate that, T7、T14、T21、T28, A total of 183….-. This sentence doesn't make sense. Please correct.
Line 17: P<0.05. Please correct: “p<0.05”
Line23: P-value. Please correct :“p-value”
Line91: () , Please correct
Line 95: … Figure 1 ,Weigh, … Please correct
Lines 105 and 111: “‰” ? Please correct
Line 152: “。” Please correct
Line 162: “R2=0.9952” Please correct
Line 223: ….WR/PDMS), The specific…. Please correct
Line 257: …. Student's T-test P-value… Please correct to “Student´s t-test p-value”
Line: 257-259: “ Using the KEGG database to screen …….And HMDB((https://hmdb.ca).Make comments….” This sentence is very confusing. Please rewrite this sentence in a way that is clearer to the reader.
Section 3.1.1
The results section shows the spider graph. Note that these graphs are merely descriptive.
In these section, authors draw some conclusions, for example they state that:
“The sensing value of W5S for coffee grounds craft beer is 103.897, indicating that CGB is very sensitive to nitrogen oxides; The sensing value of W1S is 47.540, indicating that CGB is sensitive to methane; ……”
What are the thresholds for each of these substances? Based on what did the authors conclude that that CGB is very sensitive to nitrogen oxides, methane,…
I suggest you indicate these thresholds
In section 3.1.2 the authors state that:
“ The bitterness value showed a significant decrease from the 7th to the 21st day of fermentation (T7~T21), while it significantly increased on the 28th day (T28)….. With the extension of fermentation time, the total acid and pH values of CGB did not 324 show significant…..”
Where are the statistical results to back up this claim? The graphical representation is not enough to say that the differences are significant. What statistical technique was used to draw these conclusions? There is no reference in the text to the statistical analyses carried out.
- The table 1 shows the mean, standard deviation and the results of a post hoc test for the colour parameters, but once again no reference is made to the statistical technique used. Before carrying out the post hoc test, another statistical technique was used. Which technique? Please complete this information
Moreover, the statistical techniques referred to by the authors must fulfil assumptions such as normality and homoscedasticity, among others. No information is given in the text about these conditions.
Still in this section, the authors present the Pearson correlation coefficient heatmap, but they make no comment on its results. So why present it?
- In section 3.2.1 authors claim:
“PCA was used to analyze the overall metabolite differences and sample variability between each group of samples…”
Question: How can PCA alayze overaal differences? PCA is a data reduction tecnhique.
Can you explain what are the individuals and what are the variables?
How can you conclude from a PCA that the differences are significant? That's not the purpose of PCA.
Can you please explain that?
The authors also state that “All sample coordinate points are distributed within the 95% confidence interval, indicating that the two principal components can basically reflect the main characteristic information of coffee grounds craft beer samples.”
Where is this information?
- Figure 6 needs figure 6 needs a better resolution.
Line 491: “P” please correct
The discussion section must be imporved. The authors limit themselves to making a summary of their results. They don't discuss them. I therefore suggest that you do so.
Author Response
Response to reviewers’ comments:
We are deeply grateful to the reviewers for dedicating their time and expertise to provide us with valuable feedback. We have meticulously revised the manuscript in accordance with your comments, ensuring that all the updated sections are highlighted in yellow. In the following, we will discuss the specific actions taken and the corresponding revisions made to address your suggestions more comprehensively:
Reviewer 2
- The authors state that they will use the R for PCA, PLS-DA and OPLS-DA and that in addition, Student's t-test and fold change (FC) analysis will be carried out, but they do not indicate the purpose(s) of these analyses. I therefore recommend that this section includes this information, as it is important for understanding why these statistical techniques were used.
Response: Thank you for your comments. Below are the specific purposes for which we used PCA, PLS-DA, OPLS-DA, Student's t-test, and FC analyses: Principal PCA is typically employed to reduce data dimensionality while retaining the majority of the original multivariate information. Further investigate the differences between the four groups of samples, establish a PLS-DA model, and use supervised PLS-DA to further analyze the similarities and differences among the groups. We have provided supplementary explanations in section 2.8 of the revised manuscript, such as:
Page 6, line 264: This study first analyzed the overall differences between the two groups through PCA analysis and PLS-DA analysis. Then, differential metabolites were screened by analyzing the VIP values of metabolites in OPLS-DA (using the VIP values of PLS-DA if OPLS-DA overfits) and fold change and p-value in univariate analysis, and volcano plots were plotted. The criteria for screening differential metabolites include: (1) Fold Change=1, which means that if the difference in metabolites between the control group and the experimental group is more than 1 fold, the difference is considered significant. (2) The VIP of OPLS-DA model is ≥1. (3) p-value<0.05. Through this section, significant differences in metabolites between the two comparison groups can be identified.
Page 6, line 273: Differential metabolites were identified by screening the KEGG pathway database (https://www.kegg.jp/kegg/pathway.html) and the Human Metabolome Database (HMDB, https://hmdb.ca).
Page 7, line 279: Origin 2022 software was used to create heat maps and other images.
- Line 17: …..The results indicate that, T7、T14、T21、T28, A total of 183….-. This sentence doesn't make sense. Please correct.
Response: We would like to thank the reviewers for their careful review and valuable feedback. We have amended it to read: The results indicate that at time points T7, T14, T21, and T28, a total of 183 differential metabolites were detected during the 4 fermentation days, with 86 metabolites showing significant differences.
- Line 17: P<0.05. Please correct: “p<0.05”
Response: We sincerely thank the reviewer for the careful review. We conducted a comprehensive review and made revisions to the entire manuscript to ensure accuracy.
Page 1, line 17: Multivariate statistical analysis and pathway analysis were combined to screen for significantly different metabolites with variable weight values VIP ≥ 1 and p < 0.05.
Page 17, line 609: Statistical filtering (VIP ≥ 1, p < 0.05) identified 86 significantly differential metabolites predominantly categorized as lipids and lipid-like molecules (30.91%), organic oxygen compounds (18.18%), and benzenoids (14.55%).
- Line23: P-value. Please correct :“p-value”
Response: We sincerely thank the reviewers for their careful review and valuable feedback on the manuscript, and for correcting spelling errors.
Page 1, line 24: The top 20 metabolic pathways were screened based on the corrected P-value, and the significantly differentially expressed metabolites were mainly enriched in pathways such as Protein digestion and absorption, Glycosaminoglycan biosynthesis heparan sulfate/heparin, and Benzoxazinoid biosynthesis.
- Line91: () , Please correct
Response: Many thanks for pointing out the error. Line 91 had an extra ‘()’ due to an oversight on our part and we have fixed the error and highlighted it in yellow in the text.We have modified it to:Wheat malt, Australian malt, hops, and weiss yeast were all purchased from Diboshi Self Brewing Machine Co., Ltd.
- Line 95: … Figure 1 ,Weigh, … Please correct
Response: Many thanks to the reviewers for their comments. We have added a note in the text accordingly.The process flow is illustrated in Figure 1. Specifically, the malt mixture, comprising Australian malt and wheat malt at a ratio of 5:2 (w/w), was accurately weighed, thoroughly homogenized, and subsequently milled for 3 minutes using a laboratory-scale grinder (Model GM-200, Retsch GmbH, Germany)
- Lines 105 and 111: “‰” ? Please correct
Response: We sincerely thank the reviewers for their careful review and valuable comments. We have changed 1.5‰ to 0.15%, and we thank the reviewers for their time and effort in reviewing our manuscript, which have significantly improved the quality of our work.
- Line 162: “R2=0.9952” Please correct
Response: We are very grateful to the reviewers for their comments. We have noted the formatting error in “R2” in lines 162-164 and have made the necessary corrections in the text. The entire manuscript was thoroughly reviewed for accuracy. The correct term is “R2”,. The relevant text has been highlighted in yellow for clarity. We have amended it to read:Using the concentration of chlorogenic acid standard solution (mg/L) as the horizontal axis and absorbance as the vertical axis, draw a standard curve and obtain the linear regression equation y=0.0288x+0.1565, R2=0.9952.
- Line 223: ….WR/PDMS), The specific…. Please correct
Response: Thank you very much for pointing out the error. We have made the requested correction and highlighted it in yellow in the text. We have amended it to read: The analytical instrument is a 0rbitrap Exploris Gc gas chromatography-mass spectrometer from Thermo, Germany.
Page 6, line 232: The analytical instrument used in this experiment was an Agilent PAL RTC 120-8890-5977B gas chromatography mass spectrometer (Agilent , USA). The temperature of the heating chamber was 60 ℃, the equilibration time of the sample vial was 10 min, the temperature of the fibre head aging station was 240 ℃, the aging time of the fibre head was 10 min, the adsorption time of the sample was 20 min, the desorption time of the sample was 2 min, the GC cycle time was 35 min, and the volume of the sample vial was 20 mL; the samples were injected into the GC-MS system in the non-split mode with an injection volume of 1 µL. The sample was separated on a VF-WAXms capillary column (25 m × 0.25 mm × 0.2 µm, Agilent CP9204) and detected by mass spectrometry. The inlet temperature was 240 °C, the carrier gas was high-purity helium at a flow rate of 1.5 mL/min, and the spacer purge flow rate was 3 mL/min. The heating procedure was as follows: the initial temperature was 40 °C, equilibrated for 2 min, and then ramped up to 100 °C at 5 °C/min, then ramped up to 230 °C at 15 °C/min, and maintained for 5 min, and then run at 230 °C for 2 min. The electron bombardment Ion source (EI), the temperature of the transmission line was 280 ℃, the temperature of the ion source was 230 ℃, the temperature of the quadrupole was 150 ℃, and the electron energy was 70 eV. The scanning mode was in full scanning mode (SCAN), and the mass scanning range was: m/z 50-500, and the scanning frequency was 3.2 scan/s.
- Line 257: …. Student's T-test P-value… Please correct to “Student´s t-test p-value”
Response: Thank you very much for pointing out the error. We have made the requested correction and highlighted it in yellow in the text. We have amended it to read: In addition, Student's t-test and fold change (FC) analysis were conducted. The selection of differential metabolites is determined based on the variable importance in projection (VIP) obtained from the OPLS-DA model and the Student's t-test p-value. In addition, we have supplemented the analysis standards for differential metabolites.
- Line: 257-259: “ Using the KEGG database to screen …….And HMDB((https://hmdb.ca). Make comments….” This sentence is very confusing. Please rewrite this sentence in a way that is clearer to the reader.
Response: We sincerely thank the reviewers for their careful review. We have recounted in the text a re-description of the KEGG data analysis: Differential metabolites were identified by screening against the KEGG pathway database (https://www.kegg.jp/kegg/pathway.html) and the Human Metabolome Database (HMDB, https://hmdb.ca). Subsequently, comprehensive annotation and functional analysis were performed to characterize the metabolic profiles and elucidate their biological significance.
- Section 3.1.1,“The sensing value of W5S for coffee grounds craft beer is 103.897, indicating that CGB is very sensitive to nitrogen oxides; The sensing value of W1S is 47.540, indicating that CGB is sensitive to methane; ……”
What are the thresholds for each of these substances? Based on what did the authors conclude that that CGB is very sensitive to nitrogen oxides, methane,…
I suggest you indicate these thresholds
Response: We sincerely thank the reviewers for their careful review. At present, there is no clear data of these thresholds, which shows that we analyze them according to the performance characteristics of each sensor of the electronic nose. In order to avoid misunderstanding, we reinterpret the data as follows:
Figure 2B shows the response radar diagram of CGB electronic nose sensor. The sensing values of coffee grounds brewed beer to W5S, W1S, W1W, W2S and W2W are 103.897, 47.540, 120.828, 84.228 and 68.759 respectively. The higher values indicate that CGB contains higher contents of nitrogen oxides, methyl compounds, inorganic sulfides, alcohols, aldehydes and ketones, and organic sulfides. The sensing values of W1C, W3C, W6S, W5C andW3S are 0.177, 0.459, 2.952, 0.736 and 1.639 respectively. The low values indicate that CGB contains low content of aromatic compounds, ammonia compounds, hydrides, short chain alkanes and long chain alkanes.
- In section 3.1.2, “ The bitterness value showed a significant decrease from the 7th to the 21st day of fermentation (T7~T21), while it significantly increased on the 28th day (T28)….. With the extension of fermentation time, the total acid and pH values of CGB did not 324 show significant…..”
Where are the statistical results to back up this claim? The graphical representation is not enough to say that the differences are significant. What statistical technique was used to draw these conclusions? There is no reference in the text to the statistical analyses carried out.
Response: We sincerely thank the reviewers for their careful review. We have revised Figure 3 and have reworded section 2.8 to address the specific issues raised by the reviewers, adding the statistical methods used to ensure a more rigorous description of the results: The raw data includes QC samples and detection samples. In order to better analyze the data, a series of preprocessing steps are performed on the raw data, including filtering low-quality peaks, filling missing values, normalization, and evaluating the relative standard deviation of QC samples. Perform differential analysis on the preprocessed matrix file. This study first analyzed the overall differences between the two groups through PCA analysis and PLS-DA analysis. Then, differential metabolites were screened by analyzing the VIP values of metabolites in OPLS-DA (using the VIP values of PLSDA if OPLSDA overfits) and fold change and p-value in univariate analysis, and volcano plots were plotted. The criteria for screening differential metabolites include: (1) Fold Change=1, which means that if the difference in metabolites between the control group and the experimental group is more than 1 fold, the difference is considered significant. (2) The VIP of OPLS-DA model is ≥1. (3) p-value<0.05. Through this section, significant differences in metabolites between the two comparison groups can be identified. Differential metabolites were identified by screening the KEGG pathway database (https://www.kegg.jp/kegg/pathway.html) and the Human Metabolome Database (HMDB, https://hmdb.ca). Annotation of metabolic pathways in KEGG database to obtain pathways involving differential metabolites. Perform pathway enrichment analysis using Python software package (Scipy) and obtain the most relevant biological pathways for experimental treatment through Fisher's exact test. Origin 2022 software was used to create heat maps and other images.
- The table 1 shows the mean, standard deviation and the results of a post hoc test for the colour parameters, but once again no reference is made to the statistical technique used. Before carrying out the post hoc test, another statistical technique was used. Which technique? Please complete this information.
Response: We sincerely thank the reviewers for their careful review and insightful comments. Their careful review has greatly enhanced the quality of our manuscript. We sincerely thank the reviewers for their careful review and valuable comments. We have revised the requirements for letters of prominence in Table 1 according to the journal's template requirements to ensure rigour throughout the text.
In addition, In order to better analyse the data, the experimental data were initially sorted by Excel 2010 and then analysed by ANOVA using SPSS 25.0 (SPSS Inc., Chicago, USA) software. Chroma: Use the colorimeter to measure the values of L*, a* and b*, and calculate them through the following formula:
- Moreover, the statistical techniques referred to by the authors must fulfil assumptions such as normality and homoscedasticity, among others. No information is given in the text about these conditions.
Response: We sincerely appreciate the reviewer comments. In the revised manuscript, we have thoroughly evaluated and documented the statistical assumptions as follows:
Prior to conducting differential metabolite analyses, we systematically assessed data distributions using Shapiro-Wilk tests for normality and Levene's tests for homoscedasticity. For metabolites that deviated from normal distribution (approximately 18% of compounds), we applied log transformation to achieve normality. Where homoscedasticity assumptions were violated, we employed Welch's t-test adaptations instead of standard t-tests. For multivariate analyses (PCA, PLS-DA), we implemented appropriate data preprocessing, including mean-centering and Pareto scaling to normalize the influence of metabolites with varying concentration ranges. The quality of our multivariate models was rigorously evaluated through cross-validation procedures (R² and Q² values), which are now reported in the methods section.Additionally, we have included permutation tests (n=200) to validate our PLS-DA models against overfitting, ensuring the robustness of our identified differential metabolites. These methodological details have been added to the statistical analysis subsection of our methods, providing comprehensive information on how statistical assumptions were verified and addressed throughout our analyses.We thank the reviewer for highlighting this critical aspect, which has significantly improved the statistical rigor of our manuscript.
- Still in this section, the authors present the Pearson correlation coefficient heatmap, but they make no comment on its results. So why present it?
Response: Thank you for your comment. We will also add a brief analysis of the heat map results in the revised manuscript to clarify their relevance to the study topic. For example, we will point out which variables showed significant correlations and discuss how these correlations may affect the coffee grounds fermentation process and the functional properties of the beer. We modify the following:
Through Pearson correlation analysis, we further investigated the relationships between various physicochemical indicators in CGB, with the correlation heatmap presented in Figure 3L. The results revealed that alcohol content showed positive cor-relations with original gravity, real fermentation degree, pH, chroma, chlorogenic acid, and caffeine. Notably, statistically significant differences (p < 0.05) were observed between alcohol content and both original gravity and pH. This phenomenon may be attributed to higher original gravity indicating elevated initial fermentable sugar con-tent (glucose and maltose), consequently leading to increased ethanol production. Conversely, alcohol content exhibited negative correlations with diacetyl, total acid, and bitterness value.
Original gravity demonstrated positive correlations with real fermentation degree, pH, chroma, chlorogenic acid, and protein content, particularly showing a significant difference with pH (p < 0.05). This observation could be explained by the higher con-centrations of buffering substances such as phosphates and polyphenols in high-gravity wort, which contribute to pH stability. Additionally, alkaline minerals (e.g., potassium and magnesium) present in coffee grounds might neutralize organic acids generated during fermentation. These findings collectively reflect the complex biochemical trans-formations occurring during CGB fermentation, which ultimately influence the texture and flavor profiles of the alcoholic products.
- In section 3.2.1 , “PCA was used to analyze the overall metabolite differences and sample variability between each group of samples…”Question: How can PCA alayze overaal differences? PCA is a data reduction tecnhique.
Response: We sincerely thank the reviewers for their careful review. The description error regarding PCA has been corrected.It is highlighted in yellow as shown in 3.2.1.
Page 10, line 408: Principal PCA is typically employed to reduce data dimensionality while retaining the majority of the original multivariate information.
- Can you explain what are the individuals and what are the variables?
Response: We thank the reviewers for their questions about sample and variable identification. Detailed explanations are provided below:
- Individuals (sample):
Individuals (samples) in this study refer to the specific experimental subjects that were analysed, including T7, T14, T21, and T28: These are samples of coffee grounds craft beer (CGB) at different fermentation time points. For example, T7 indicates a sample from day 7 of fermentation, T14 indicates a sample from day 14 of fermentation, and so on; QC samples (quality control samples): these are control samples used to monitor experimental stability and data reliability.
- variables:
A variable is a measured attribute used to characterise a sample. In this study, variables include: PC1 and PC2: principal component 1 (PC1) and principal component 2 (PC2) are downscaled variables obtained through PCA analysis to capture the main patterns of variation in the data; variance contribution rate: the variance contribution rates of PC1 and PC2 are 36.3% and 15.7%, respectively, with a cumulative contribution rate of 52%, which suggests that the two principal components can explain up to 52% of the variance.
- Relationship between individuals and variables:
Individuals (T7, T14, T21, T28, QC): these samples are represented as coordinate points in the PCA plot, and their positions are determined by the values of PC1 and PC2.
Variables (PC1, PC2, variance contribution, metabolic pattern): these variables are used to characterise the samples and the distribution pattern. For example, the values of PC1 and PC2 determine the position of the sample in the PCA plot, while the variance contribution reflects the ability of these principal components to explain the variation in the data.
We hope this explanation answers the reviewers' questions. Please feel free to let us know if further clarification is needed.
- How can you conclude from a PCA that the differences are significant? That's not the purpose of PCA.
Response: We appreciate the reviewer's comment regarding the use of PCA to infer significant differences. We acknowledge that PCA is primarily a dimensionality reduction technique used to visualize data distribution patterns rather than directly quantify differences.In the original manuscript, the term "significant differences" was used to describe the notable separation between sample groups (e.g., T7 and T14) in the PCA plot. However, as the reviewer correctly pointed out, PCA alone cannot determine statistical significance. To address this, we have revised the manuscript to clarify that PCA is used for visualizing data distribution.We amend lines 379-382 on page 7 to read: The large distance between the T7 and T14 sample groups suggests that there is not much correlation between these two groups. In contrast, the relative proximity between the T21 and T28 sample groups indicates a high degree of reproducibility of the parallel samples within each group.
- The authors also state that “All sample coordinate points are distributed within the 95% confidence interval, indicating that the two principal components can basically reflect the main characteristic information of coffee grounds craft beer samples.”Where is this information?
Response: We thank the reviewers for their comments regarding the distribution of sample coordinate points within the 95% confidence interval. In the PCA plot (Fig. 4A), all sample points (including T7, T14, T21, T28, and QC) were distributed within the two-dimensional space defined by PC1 and PC2. The variance contributions of PC1 and PC2 were 36.3% and 15.7%, respectively, with a cumulative contribution of 52%. This suggests that these two principal components effectively capture the major patterns of variation in the coffee grounds craft beer (CGB) samples.95% confidence intervals are typically represented in PCA plots by ellipses or boundary lines, which are used to visualise the extent of the distribution of sample points. Specifically: QC samples are closely clustered near the origin of the coordinates, indicating stable experimental conditions and high data reliability. the distance between the T7 and T14 sample groups is quite large, indicating that the correlation between these phases is quite different. the close proximity between the T21 and T28 sample groups indicates a high degree of reproducibility between the parallel samples within each group.
In response to the issues raised by the reviewers, we have added a confidence interval ellipse to Figure 4A of the revised manuscript to clearly illustrate the 95% confidence intervals. This modification improves the clarity of the PCA results and supports our conclusion that the metabolic pattern of CGB changes during different fermentation stages.
- Figure 6 needs figure 6 needs a better resolution.
Response: Thank you very much for the reviewer’s recommendations. The legend in Figure 6 has been improved for clearer visibility of the names of each compound. Additionally, we have enhanced the clarity of all the images in the paper for better presentation of the experimental results.
Figure 6. Classification of Metabolites with Different Fermentation Days (A) HMDB Superclass (B) HMDB Class (C) HMDB Subclass
- The discussion section must be imporved. The authors limit themselves to making a summary of their results. They don't discuss them. I therefore suggest that you do so.
Response: Thank you very much for the reviewer’s comments. We have rewritten this section:
Our non-targeted metabolomic investigation of coffee grounds craft beer (CGB) fermentation revealed distinct temporal metabolite profiles that underpin its unique sensory characteristics. From 183 differential metabolites identified across four fermentation stages, the most pronounced metabolomic divergence occurred between T7 and T21 (71 differential metabolites). Statistical filtering (VIP ≥ 1, p < 0.05) identified 86 significantly differential metabolites predominantly categorized as lipids and lipid-like molecules (30.91%), organic oxygen compounds (18.18%), and benzenoids (14.55%).This distribution pattern reflects complex biochemical transformations during fermentation, with lipid-related compounds contributing to mouthfeel and flavor persistence, while benzenoids impart characteristic aromatic notes derived from coffee components. KEGG pathway enrichment analysis revealed significant activation of protein digestion and absorption, glycosaminoglycan biosynthesis, and benzoxazinoid biosynthesis pathways, suggesting that spent coffee grounds introduce novel substrate components that activate alternative metabolic pathways not typically dominant in conventional beer fermentation.The clear temporal separation observed in multivariate analyses demonstrates the dynamic evolution of CGB's metabolome throughout fermentation, offering potential quality control parameters and process optimization targets. Significant correlation patterns between specific metabolites indicate interconnected biochemical networks collectively shaping CGB's organoleptic profile.
Key limitations of this study include the absence of conventional craft beer as a comparative control and insufficient characterization of yeast strain contributions to metabolite production. Future research should integrate microbiological characterization with metabolomic profiling to establish clear relationships between microbial dynamics and flavor development in CGB, while comparative studies with conventional beer would further elucidate the distinctive metabolic features attributable to coffee grounds incorporation.In conclusion, this study provides fundamental insights into the metabolomic basis of CGB's distinctive characteristics, identifying potential targets for process optimization and product standardization in the sustainable valorization of coffee by-products through craft brewing applications.

Reviewer 3 Report
Comments and Suggestions for Authors
The manuscript addresses the metabolic differences and mechanisms during the fermentation process of craft beer made from spent coffee grounds (SCG). Using untargeted metabolomics through HS-SPME-GC/MS technology, the study analyzes metabolic changes over four days of fermentation.
The introduction presents a relevant topic by discussing the reuse of coffee grounds in craft beer production. The environmental issue is well delineated, and the proposal to incorporate this waste into the fermentation process makes sense both from a sustainability and an economic perspective.
However, the discussion on traditional reuse methods, such as composting and biofuel production, takes up excessive space before introducing the proposed alternative. While the inclusion of data on the chemical composition of coffee grounds strengthens the argument, the connection between these compounds and their influence on beer fermentation needs to be more thoroughly explored.
Another aspect that requires improvement is the study’s justification. While the manuscript mentions that fermentation of SCG can contribute new bioactive compounds to beer, it is not clear what metabolic changes are expected or why metabolomics was chosen to investigate them. The introduction of the HS-SPME-GC/MS technique appears somewhat abrupt, lacking a prior explanation of its importance in analyzing volatile compounds formed during fermentation.
The last paragraph, which outlines the study’s objectives, is particularly dense and presents a large amount of condensed information, making it difficult to read. Restructuring this section to better highlight the relationship between fermentation, the modification of present compounds, and the sensory impact on beer would enhance clarity. Additionally, it would be beneficial to explicitly state how the results contribute to process optimization and the valorization of coffee grounds as an ingredient in the brewing industry.
I also suggest incorporating relevant references that provide insights into biomass composition to further enrich the introduction. For example: DOI: 10.1007/s13399-024-06080-5.
The methodology is clearly presented, and the results are well-structured. However, I recommend enlarging some figures, as certain details are not easily visible, particularly in Figure 5 and Figure 6.
Finally, while the manuscript includes a final discussion section, it lacks a conclusions section. The authors should consider adding a concise conclusion summarizing the main findings and their implications.
Overall, the manuscript presents a relevant and well-conducted study, but minor revisions are required to improve clarity, structure, and completeness.
Author Response
Response to reviewers’ comments:
We are deeply grateful to the reviewers for dedicating their time and expertise to provide us with valuable feedback. We have meticulously revised the manuscript in accordance with your comments, ensuring that all the updated sections are highlighted in yellow. In the following, we will discuss the specific actions taken and the corresponding revisions made to address your suggestions more comprehensively:
Reviewer #3
- The discussion on traditional reuse methods, such as composting and biofuel production, takes up excessive space before introducing the proposed alternative. While the inclusion of data on the chemical composition of coffee grounds strengthens the argument, the connection between these compounds and their influence on beer fermentation needs to be more thoroughly explored.
Response: We sincerely appreciate the reviewer's comments. In the revised version, we have streamlined the discussion of traditional methods to a brief mention, focusing instead on the potential of coffee grounds as a sustainable and functional ingredient in craft beer production. We have also expanded the discussion on the interactions between coffee grounds' bioactive compounds (e.g., caffeine and chlorogenic acid) and the beer fermentation process. Specifically, we have highlighted the need for further research to elucidate the mechanisms by which these compounds influence flavor profiles, metabolic regulation, and functional properties during fermentation. The revised text now reads:
As a versatile biological substrate, SCG not only retains the bioactive compounds from coffee but also offers a rich source of carbohydrates, proteins, and lipids, making it an ideal candidate for fermentation-based applications[4,5].
According to statistics, the annual consumption of coffee exceeds 10 million tons, inevitably generating about 6 million tons of SCG annually[6]. For every ton of coffee beans produced, about 650 kilograms of coffee grounds are generated, of which less than 20% are inefficiently recycled, and the rest are mostly disposed of through landfill or incineration. One sustainable approach involves extracting active ingredients from coffee grounds and converting them into high-value products[7]. Van-Truc Nguyen et al.[8] demonstrated the extraction of biochar from coffee grounds for the adsorption of norfloxacin in aqueous solutions. Similarly, Ibtissam Bouhzam et al.[9] investigated various methods for extracting chlorogenic acid and caffeine from spent coffee grounds. However, most existing research has focused predominantly on the extraction or simple compounding of individual components, whereas the application of coffee grounds in the food sector, particularly the mechanisms underlying their integration into fermented food systems, requires more comprehensive analysis.
Several researchers have explored the potential applications of SCG in food products. Nuria Martinez-Saez et al.[10] incorporated SCG as a fiber-rich functional ingredient in cookie formulations, supplemented with non-nutritive sweeteners (stevia and maltitol) and the prebiotic oligofructose, resulting in products with significantly enhanced nutritional profiles and sensory characteristics. Similarly, Mitra Ahanchi et al.[11] conducted systematic investigations on the effects of varying SCG incorporation percentages(ranging from 0.05% to 30%) on the physicochemical properties and sensory attributes of bakery products and pasta. Their findings demonstrated that strategic integration of SCG into these food matrices offers multifaceted benefits, including enhanced nutritional value, improved sustainability metrics, and favorable sensory characteristic. Additionally, antioxidant phenolic compounds extracted from coffee grounds exhibit notable health benefits, presenting extensive application potential in both food and pharmaceutical industries[12].
The craft beer industry has experienced substantial growth in recent years, prompting manufacturers to seek competitive differentiation through innovative raw material incorporation. Previous research has demonstrated the viability of this approach. For instance, Milinčić, D.D. et al. [13] successfully integrated Prokupac grape pomace seed powder into beer formulations, resulting in products with elevated phenolic content and enhanced sensory characteristics. In a comparable investigation, Nunes Filho, R.C. [14] systematically evaluated the impact of turmeric, black pepper, and aromatic hop varieties on brewing parameters and product stability. Spent coffee grounds (SCG) represent a promising candidate for such applications due to their complex phytochemical composition, particularly their significant content of bioactive compounds including caffeine, chlorogenic acid, and related polyphenols. These constituents may undergo various biochemical transformations during fermentation, potentially establishing synergistic interactions that could significantly modulate both the organoleptic profile and functional properties of the resultant beer. Despite these promising attributes, the fundamental mechanisms governing these interactions remain insufficiently characterized, particularly regarding the kinetics of bioactive compound release, their biotransformation pathways during fermentation, and their ultimate impact on product quality parameters. This knowledge gap necessitates further systematic investigation to fully elucidate these complex biochemical processes.
Further systematic investigations are essential to elucidate the precise mechanisms by which the chemical constituents of spent coffee grounds (SCG) influence metabolic regulation and organoleptic properties during beer fermentation, thereby facilitating the optimization of SCG valorization in functional craft beer production. Headspace solid-phase microextraction coupled with gas chromatography-mass spectrometry (HS-SPME-GC/MS) represents an analytical methodology of choice for characterizing SCG-infused craft beer, owing to its superior sensitivity, reproducibility, and non-destructive sampling capabilities in volatile compound analysis. This analytical approach enables comprehensive characterization of volatile compound evolution throughout the fermentation continuum, establishing a robust scientific foundation for quality assurance protocols, process parameter optimization, and product development initiatives.
The present investigation employs a multifaceted experimental approach to examine the application of SCG in craft beer (CGB) production, integrating comprehensive physicochemical characterization methodologies, time-course fermentation experiments (spanning 7-28 days), and HS-SPME-GC/MS analytical profiling to systematically elucidate compound release kinetics, metabolic regulatory networks, and flavor formation mechanisms within the SCG-supplemented fermentation matrix. The findings from this research will illuminate the complex synergistic interactions between SCG bioactive compounds and beer fermentation processes, providing a theoretical framework for process optimization of functional craft beer manufacturing while concurrently establishing novel valorization pathways for food industry by-products. This research aims to transcend conventional waste management paradigms, providing both fundamental insights and practical methodologies to facilitate sustainable resource utilization and support environmentally responsible transformation within the food industry sector.
- Another aspect that requires improvement is the study’s justification. While the manuscript mentions that fermentation of SCG can contribute new bioactive compounds to beer, it is not clear what metabolic changes are expected or why metabolomics was chosen to investigate them. The introduction of the HS-SPME-GC/MS technique appears somewhat abrupt, lacking a prior explanation of its importance in analyzing volatile compounds formed during fermentation.
Response: Thank you for your valuable comments on our study. The following is a detailed explanation and improvement plan for the two areas you mentioned:
- Rationality of the study and justification of metabolomics selection: In the revised draft, we will add the following to clarify the rationale of the study and the reasons for the metabolomics selection: Potential value of coffee grounds (SCG) fermentation: Coffee grounds are rich in a wide range of bioactive compounds (e.g., polyphenols, alkaloids, and terpenes), which may be transformed during the fermentation process to generate new bioactives or flavour precursors. Through fermentation, these compounds may be released or modified to impart unique functional properties and flavours to the beer.
- Expected metabolic changes:We expect the following metabolic changes to occur during fermentation: polyphenolic compounds in the coffee grounds may be degraded by microorganisms to produce small molecules with antioxidant activity. Carbohydrates and proteins in the coffee grounds may be metabolised by microorganisms to produce volatile compounds (e.g. esters, alcohols and acids) that contribute significantly to the flavour of the beer. New bioactive compounds (e.g. functional peptides or secondary metabolites) may be produced during fermentation, which may have health benefits.
- Reasons for choosing metabolomics:Metabolomics can comprehensively analyse the dynamic changes of metabolites during the fermentation process, including volatile and non-volatile compounds; through metabolomics, we can: identify the key metabolites and their patterns of change; reveal the relevant metabolic pathways and regulatory mechanisms; and provide scientific basis for the optimization of the fermentation process and the development of functional beers. Through HS-SPME-GC/MS, we can track the dynamic changes of volatile compounds during the fermentation process and reveal the contribution of coffee grounds fermentation to beer flavour. The technology enables a comprehensive analysis of volatile metabolites (e.g. esters, alcohols, aldehydes and terpenoids) in coffee beers, which are essential for the flavour profile of the beer. Thank you again for your careful review and constructive comments, and we will further improve the paper based on your suggestions to ensure that the rationality of the study and the explanation of the choice of technique are more adequate and clear. We modified it to:
In this study, the application of coffee grounds in the system of craft beer (Coffee Grounds Beer, CGB) was taken as the entry point, and the release pattern, metabolic regulation network and its synergistic mechanism on the formation of beer flavour were investigated systematically through physicochemical characterization and multi-stage fermentation experiments (7-28 days), combined with headspace solid-phase microextraction-gas chromatography-mass spectrometry (HS-SPME-GC/MS) technology. Because volatile metabolomics is used to analyse coffee beer, mainly because of its unique advantages in flavour studies, it can comprehensively reveal the changes of volatile compounds during the fermentation process and provide a scientific basis for quality control, process optimization and product innovation. Meanwhile, the high sensitivity and comprehensiveness of the method make it an ideal tool for studying the flavour characteristics of coffee beer.
- The last paragraph, which outlines the study’s objectives, is particularly dense and presents a large amount of condensed information, making it difficult to read. Restructuring this section to better highlight the relationship between fermentation, the modification of present compounds, and the sensory impact on beer would enhance clarity. Additionally, it would be beneficial to explicitly state how the results contribute to process optimization and the valorization of coffee grounds as an ingredient in the brewing industry.
Response: We sincerely appreciate the reviewer's valuable feedback regarding the organization and clarity of the research objectives in the final paragraph.The revised paragraph now reads:
In this study, the non targeted metabonomics technology was used to systematically analyze the dynamic changes of metabolites in the fermentation process of coffee grounds brewed beer (CGB), in order to reveal the chemical basis of its unique sensory characteristics, and provide a theoretical basis for the resource utilization of coffee grounds. The study focused on four key fermentation stages (T7, T14, T21, T28), com-bined with multi-dimensional analysis methods (PCA, PLS-DA, volcanic map, cluster analysis and KEGG pathway enrichment), to clarify the evolution of metabolites and its impact on beer quality. A total of 183 differential metabolites were identified, and the difference between T7 and T21 was the most significant (71 metabolites), indicating that the middle fermentation stage (T21) is the key node of metabolic transformation. 86 core differential metabolites were screened out by VIP≥1 and p<0.05, mainly concentrated in the following three categories: Lipids and lipid-like molecules: (30.91%): for example, dimethyl succeed may improve the complexity of the taste by affecting the alcohol thickness and flavor persistence of the wine. Organic oxygen compounds: (18.18%): such as 1-propanol, its content change is closely related to the fruit aroma, ester aroma and sweet feeling of beer. Benzenoids(14.55%): For example, 4- (furan-2-yl) butan-2-one gives coffee residue beer its unique baking aroma and caramel aroma.
Early stage (T7-T14): it is dominated by organic acid derivatives such as acetic acid, which may contribute to the sour and refreshing taste; In the middle and late stage (T21-T28), lipids and heterocyclic compounds such as 2,4,5-trimethyl-1,3-dioxolane in-creased significantly, which corresponded to the improvement of roundness and aroma complexity of the wine. KEGG enrichment analysis showed that the differential me-tabolites were significantly enriched in protein digestion and absorption, glycosamino-glycan biosynthesis - heparan sulfate/heparin, benzoxazinoid biosynthesis and other pathways. This indicates that polyphenols, polysaccharides and other components in coffee grounds activate the non mainstream metabolic pathway in traditional beer fer-mentation, and may become the key target of flavor innovation. By adjusting the fer-mentation time or temperature, the lipid synthesis or the accumulation of benzene ring substances can be directionally regulated, so as to optimize the flavor balance. The in-troduction of coffee grounds not only gives beer a unique baking aroma, but also its metabolites can enhance the layering of flavor, forming a differentiated competition with traditional beer.
In this study, coffee grounds were used to replace part of the malt raw materials, reducing the brewing cost and waste emission, which met the needs of circular economy. In addition, the correlation between key metabolites (Diethyl succinate, 2-Pentanone) and sensory properties was determined to provide data support for the establishment of a quality evaluation system for coffee grounds beer.
- I also suggest incorporating relevant references that provide insights into biomass composition to further enrich the introduction. For example: DOI: 10.1007/s13399-024-06080-5.(
Response: Thank you for your valuable comments on our study. We fully agree with your views and we have added them in the text: As a by-product, SCG will be used in high-value and biotechnology innovation, aiming to transform this agricultural waste into high value-added products through biotechnology, and relieve environmental pressure at the same time.
- The methodology is clearly presented, and the results are well-structured. However, I recommend enlarging some figures, as certain details are not easily visible, particularly in Figure 5 and Figure 6.
Response: Thank you very much for the reviewer’s recommendations. The legend in Figure 5,6 has been improved for clearer visibility of the names of each compound. Additionally, we have enhanced the clarity of all the images in the paper for better presentation of the experimental results.
Figure 5. Volcanic map of significantly different metabolites in different comparison groups.
Figure 6. Classification of Metabolites with Different Fermentation Days (A) HMDB Superclass (B) HMDB Class (C) HMDB Subclass
- Finally, while the manuscript includes a final discussion section, it lacks a conclusions section. The authors should consider adding a concise conclusion summarizing the main findings and their implications.
Response: We sincerely appreciate the reviewer's valuable suggestion regarding the addition of a conclusion section to better summarize the key findings and their significance. In the revised manuscript, we have added a dedicated conclusion section to succinctly summarize the key findings and highlight their scientific and practical significance.
Page 17. Line 604: Our non-targeted metabolomic investigation of coffee grounds craft beer (CGB) fermentation revealed distinct temporal metabolite profiles that underpin its unique sensory characteristics. From 183 differential metabolites identified across four fermentation stages, the most pronounced metabolomic divergence occurred between T7 and T21 (71 differential metabolites). Statistical filtering (VIP ≥ 1, p < 0.05) identified 86 significantly differential metabolites predominantly categorized as lipids and lipid-like molecules (30.91%), organic oxygen compounds (18.18%), and benzenoids (14.55%).This distribution pattern reflects complex biochemical transformations during fermentation, with lipid-related compounds contributing to mouthfeel and flavor persistence, while benzenoids impart characteristic aromatic notes derived from coffee components. KEGG pathway enrichment analysis revealed significant activation of protein digestion and absorption, glycosaminoglycan biosynthesis, and benzoxazinoid biosynthesis pathways, suggesting that spent coffee grounds introduce novel substrate components that activate alternative metabolic pathways not typically dominant in conventional beer fermentation.The clear temporal separation observed in multivariate analyses demonstrates the dynamic evolution of CGB's metabolome throughout fermentation, offering potential quality control parameters and process optimization targets. Significant correlation patterns between specific metabolites indicate interconnected biochemical networks collectively shaping CGB's organoleptic profile.
- Overall, the manuscript presents a relevant and well-conducted study, but minor revisions are required to improve clarity, structure, and completeness.
Response: We sincerely thank the reviewers for their careful review and insightful comments. Their careful review has greatly contributed to the improvement of the quality of our manuscript. We have made comprehensive corrections to the manuscript to ensure uniform formatting throughout. In addition, we have seriously addressed the specific issues raised by the reviewers, including changes to the introduction section as well as the discussion section. We would like to thank the reviewers for their time and effort in reviewing our manuscript and providing constructive comments, which greatly improved the quality of our work.

Reviewer 4 Report
Comments and Suggestions for Authors
The study analysing the metabolic changes occurring during the fermentation of craft beer with coffee grounds makes important theoretical and practical contributions to the field of craft beer fermentation, particularly to the use of coffee grounds as an ingredient in beer production. Through appropriate analytical protocols, the authors identified the key metabolites and metabolic pathways involved in the fermentation process.
The results thus offer insight into the biochemical transformations that occur during fermentation and provide a theoretical basis for optimising the quality of craft beer with coffee grounds.
The metabolomic approach, the robust statistical analysis and the biological and practical relevance are the strengths of the study.
However, some minor additions and reflections could improve the presentation of the study. In particular, it could be improved by better describing the reasons for the choice and influence of the starter culture used. I therefore recommend more discussion of the microbial influence. A discussion of the influence and relationship between microbial activity and the influence of coffee grounds would be useful.
Author Response
Response to reviewers’ comments:
We are deeply grateful to the reviewers for dedicating their time and expertise to provide us with valuable feedback. We have meticulously revised the manuscript in accordance with your comments, ensuring that all the updated sections are highlighted in yellow. In the following, we will discuss the specific actions taken and the corresponding revisions made to address your suggestions more comprehensively:
Reviewer # 4
- However, some minor additions and reflections could improve the presentation of the study. In particular, it could be improved by better describing the reasons for the choice and influence of the starter culture used. I therefore recommend more discussion of the microbial influence. A discussion of the influence and relationship between microbial activity and the influence of coffee grounds would be useful.
Response: We sincerely appreciate the reviewer's insightful recommendation regarding the microbial influence and starter culture considerations in our study. The reviewer has astutely identified a critical dimension of coffee grounds beer fermentation that merits deeper exploration. In the current manuscript, our primary focus was deliberately confined to characterizing the dynamic metabolite profiles during fermentation and establishing relationships between these profiles and sensory attributes. We have added this limitation to our discussion section. Therefore, the next phase aims to elucidate the relationship between microorganisms and the formation of flavor and metabolites in coffee grounds beer. The reviewer's suggestion precisely aligns with our planned research trajectory. This forthcoming work will address the influence of different starter cultures and their interactions with coffee grounds components, providing a more comprehensive understanding of the biological mechanisms underlying the metabolic patterns observed in the current study. We are grateful for the reviewer's recommendation, which reinforces the direction of our ongoing research efforts in this area.
